# Developing climate-resilient rice varieties (BRRI dhan97 and BRRI dhan99) suitable for salt-stress environments in Bangladesh

**Sanjoy K. Debsharma**[1], **M. Akhlasur Rahman**[1]*, **Mahmuda Khatun**[1], **Ribed F. Disha**[1], **Nusrat Jahan**[1], **Md. Ruhul Quddus**[2], **Hasina Khatun**[1], **Sharifa S. Dipti**[3], **Md. Ibrahim**[4], **K. M. Iftekharuddaula**[1], **Md. Shahjahan Kabir**[5]

1 Plant Breeding Division, Bangladesh Rice Research Institute, Gazipur, Bangladesh, 2 Hybrid Rice Division, Bangladesh Rice Research Institute, Gazipur, Bangladesh, 3 Grain Quality and Nutrition Division, Bangladesh Rice Research Institute, Gazipur, Bangladesh, 4 Rice Farming System Division, Bangladesh Rice Research Institute, Gazipur, Bangladesh, 5 Director General, Bangladesh Rice Research Institute, Gazipur, Bangladesh

* akhlas08@gmail.com

**Data Availability Statement:** All relevant data are within the paper and its Supporting Information files.

## Abstract

Salinity variations are the main reason for rice yield fluctuations in salt-prone regions throughout the dry season (*Boro* season). Plant breeders must produce new rice varieties that are more productive, salt tolerant, and stable across a variety of settings to ensure Bangladesh's food sustainability. To assess the yield and stability, we used fifteen rice genotypes containing two tolerant checks BRRI dhan67, Binadhan-10 and the popular *Boro* rice variety BRRI dhan28 in different salinity "hotspot" in three successive years followed by additive main effects and multiplicative interaction (AMMI) model utilizing a randomized complete block (RCB) design with two replications. Parents selection was done based on estimated breeding values (EBVs). Eight parents with high EBVs (IR83484-3-B-7-1-1-1, IR87870-6-1-1-1-B, BR8992-B-18-2-26, HHZ5-DT20-DT2-DT1, HHZ12-SAL2-Y3-Y2, BR8980-B-1-3-5, BRRI dhan67, and Binadhan-10) might be used to develop new segregating breeding materials. Based on farmer preferences and grain acceptability, three genotypes (IR83484-3-B-7-1-1-1, HHZ5-DT20-DT2-DT1, and HHZ12-SAL2-Y3-Y2) were the winning and best ones. The above three genotypes in the proposed variety trial showed significantly higher yields than the respective check varieties, high salinity tolerance ability, and good grain quality parameters. Among them, HHZ5-DT20-DT2-DT1 and IR83484-3-B-7-1-1-1 harbored eight and four QTL/genes that regulate the valuable traits revealed through 20 SNP genotyping. Finally, two genotypes IR83484-3-B-7-1-1-1 and HHZ5-DT20-DT2-DT1 were released as high salinity-tolerant rice varieties BRRI dhan97 and BRRI dhan99, respectively in Bangladesh for commercial cultivation for sustaining food security and sustainability.

## 1. Introduction

One of the major cereal crops worldwide is rice (*Oryza sativa* L.), which is a predominant grain for the larger part of the world's population, especially in Asia [1, 2]. In 2018, the

**Funding:** The authors received no specific funding for this work.

**Competing interests:** The authors have declared that no competing interests exist.

worldwide rice cultivated land was 166.08 million hectares, along with a production of 769.82 million tons [3]. Rice is widely cultivated in the Asia region and accounts for 50% to 80% of people's daily caloric consumption [4, 5]. Bangladesh is the fourth-largest rice producer country in the world, producing 56.41 million tons of rice on 11.91 million hectares of land [3]. Both globally and in Bangladesh, rice production is critical to ensuring food safety. With the expectation that there would be 9.6 billion people on the planet by 2050 [6, 7]. There is a strong link between rising rice output and meet rising global food consumer demand. To feed and nourish them, yield production will need to rise by almost 70% [8, 9]. We must make the best use of all the land resources in order to increase rice production through judicious utilization and bringing the lands that are not under cultivation due to high level of salinity. This is a challenging task because it requires not just a significant increase in agricultural output but also completion in an uncertain climate [10]. According to reports, nearly one-third of the world's cultivable land is saline-prone, and more than half of all cultivable areas could be under-salinized by 2050, making salinity a big warning to sustainable agriculture production [11, 12]. The huge amount of agricultural land in Southeast and South Asian countries that were once suitable for the rice cultivation are now either not being cultivated or producing low yields due to salinity [4]. Bangladesh's salinity-affected areas cover 2.85 million hectares of land in the country's southern coastal zone [13], providing 16 percent of the country's total rice production [14]. Salt intrusions are becoming more common along this coastal belt, posing a threat to rice production and other growing crops [15]. The cultivable land is negatively devastated by salinity intrusions. Both the salt-afflicted areas and the human population are steadily expanding day by day. To feed the gradually increasing people in Bangladesh, coastal saline areas should be converted to paddy cultivation to ensure the country's food security and sustainability [16–18].

Therefore, the most pragmatic cost-effective, and environmentally beneficial strategy to address this momentous issue is to develop salt-tolerant rice varieties. Rice is a naturally salt-responsive crop and is thought to have a salinity permissible limit of 3 dS/m for soil-saturated extract (EC) [19, 20]. Apart from varying salinity intensity, location-specific meteorological conditions also have a role in genotype-environment (G-E) interaction, making it difficult to discover stable and best genotypes. With rising temperatures, the level of salinity also increases, resulting in more yield reduction [21].

The stages of rice that are most vulnerable to salinity are the seedling and flowering stages. Polygenic attributes control grain production under salinity stress, and the environmental impact is more obvious in polygenic attributes than monogenic attributes. As a result, analyzing salt tolerance polygenic features across different locations is problematic. Strong statistical analysis is necessary when genotypes are studied under various salt stress levels at various places across seasons in order to make pertinent inferences. Understanding the causes and nature of genotype-by-environment interactions (GEI) in saline-prone areas will aid in identifying the genotype that is stable in such areas [22].

GGE-biplot and AMMI model [23, 24] are two methods for evaluating G-E interactions [25]. AMMI model distinguishes environment and genotype major effects from GEI [26] and gives insight into GEI [24]. This model generates a "which-won-where" pattern, as well as winning genotype information and their adaptability.

Investigating the prevalence of beneficial alleles for various traits associated with interest linked to abiotic, biotic stresses, and grain quality traits requires the characterization of genotypes using trait-based SNPs. This chance to select genotypes utilizing cutting-edge breeding techniques, such as marker-assisted forward breeding and genomic selection, has been made possible by the high throughput SNP platform.

Breeders are always concentrating their attempts on creating elite materials that are superior to existing ones in salinity aspects, yield performance, and acceptable grain qualities. Moreover, to counteract the negative consequences of climate change, more quick and effective breeding procedures are required [27]. Due to a lack of varietal possibilities in saline-affected areas, improved, high-yielding cultivars are needed to raise national productivity.

This study's goals were to identify high and stable yield performer genotypes under salt stress with acceptable grain and cooking qualities, as well as to quantify the breeding values and reliability of the tested breeding lines. To detect high-yielder and stable rice genotypes adaptable to the vast range of stress conditions based on consumer and farmers' preferences, we examined a group of elite breeding lines for grain yield and other eating qualities in salinity 'hotspot' locations.

## 2. Materials and methods

### 2.1 Evaluation of locations

The present study was evaluated for three consecutive years in various environments in the southern coastal areas of Bangladesh. In the Regional Yield Trial (RYT), Assasuni, Kaliganj and Debhata in Satkhira district and Koyra at Khulna districts represented different salinity levels during *Boro* season 2016–17. Assasuni and Debhata are favorable, Koyra in medium, and Kaliganj in high stress based on salinity level. The seven locations (S1 = BRRI Gazipur, S2 = Debhata, S3 = Assasuni, S4 = Batiaghata, S5 = Paikgacha, S6 = Kalapara, S7 = Pathorghata) in Advanced Lines Adaptive Research Trial (ALART) and eight (S1 = Tala, S2 = Debhata, S3 = Kaliganj, S4 = Dumuria, S5 = Paikgacha, S6 = Batiaghata, S7 = Rampal, S8 = Kalapara) in Proposed Variety Trial (PVT) during three consecutive *Boro* growing seasons 2017 to 2019. The different experimental sites are shown in the geographical map in Fig 1. Table 1 reveals the salinity status and ranges of the different trials and locations across the studied areas.

### 2.2 Plant genetic materials

A total of 64 entries of salinity breeding materials were introduced in BRRI (Bangladesh Rice Research Institute) from IRRI (International Rice Research Institute) as segregating materials. After that, we conducted an observational yield trial (OYT) of these genotypes along with 21 BRRI-developed genotypes. Based on yield performances and uniformity, thirty-five genotypes were selected in the preliminary yield trial (PYT) evaluated in Assasuni, Satkhira and BRRI, Gazipur and 23 were selected in the secondary yield trial (SYT) conducted in five sites (BRRI, Gazipur; one site Khulna; three sites of Satkhira district). In RYT, best performing fifteen genotypes consisting of twelve advanced lines [(five lines from BRRI; BR8940-B-17-4-7, BR8943-B-20-9-22, BR8980-4-6-5, BR8980-B-1-3-5, BR8992-B-18-2-26); five lines from IRRI (IR86385-85-2-1-B, IR83484-3-B-7-1-1-1, IR87872-7-1-1-2-1-B, IR86385-117-1-1-B, IR87870-6-1-1-1-B)]; two lines from green super rice (GSR) genotypes (HHZ12-SAL2-Y3-Y2, HHZ5-DT20-DT2-DT1) and three check varieties (BRRI dhan28 as susceptible check, BRRI dhan67 as tolerant check and Binadhan-10 released by Bangladesh Institute of Nuclear Agriculture (BINA) was another tolerant check) were evaluated. For ALART and PVT, selected three genotypes (from RYT); IR83484-3-B-7-1-1-1, HHZ12-SAL2-Y3-Y2, and HHZ5-DT20-DT2-DT1 were evaluated in different salt-affected areas of Bangladesh. However, the variety release system in Bangladesh for BRRI developed breeding lines and exotic introduced advanced breeding materials like IR 83484-3-B-7-1-1-1 and HHZ5-DT20-DT2-DT1 is shown in Fig 2. Many genotypes having salinity tolerant QTL or genes are found diverse in terms of yield performance and level of salinity tolerance.

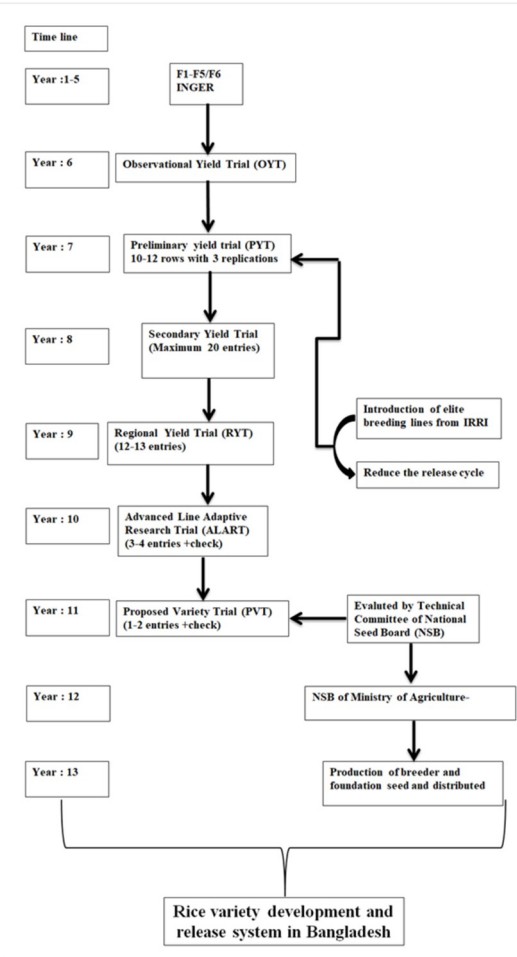

**Fig 1. Geographic illustration of the different experimental locations across Bangladesh.** The green color indicates the studied area where trials were conducted. Most studied areas were located in the southern coastal regions of Bangladesh, near the Bay of Bengal is the northeastern part of the Indian Ocean.

## 2.3 Experimental design and agronomic practices

The experiment was conducted in the dry (*Boro*) season for three consecutive years. The soaking, sowing, and transplanting of tested genotypes were executed at different experimental sites in the mentioned three successive years. Seedlings that were forty to forty-five days of age were transplanted at a 20 × 20 cm distance, with two to three seedlings per hill. The unit plot had 10 rows and a length of 5.4 meters (10.8 m$^2$). A RCB design replicated three times was used for the outfield layout. Fertilizers were supplied at a rate of 120:19:60: 20:4.01 kg NPKSZn/ha (260-97-120-110-11 kg/ha, respectively in the form of urea, TSP, MoP, gypsum, and zinc sulphate). During the last stage of land preparation, all fertilizers aside from urea were applied as a base. Urea was given in three divided doses at 10–15 days following transplantation, 4–5 days before the tillering stage, and 5–7 days through the panicle initiation stage. Crop management tasks, like irrigation and weeding, were completed on schedule. Pests, diseases, and other issues were well managed. The harvested area was 10.2 square meters. To reduce the border impacts, two consecutive border rows were removed from each of the harvested plots. Data on grain yield (tha$^{-1}$) were gathered at a moisture content of 14%. The conventional method described by [28] was used to collect all data.

**Table 1. The characterization of the studied areas including trial locations, salinity status, and salinity ranges (minimum to maximum).**

| Trials | Location name | Status of salinity | Ranges (dS/m) |
|---|---|---|---|
| RYT | Assasuni, Satkhira | Low stress (non-saline to very slightly saline) | 3.73–5.25 |
| | Kaliganj, Satkhira | High stress (very slight saline to high saline) | 3.96–15.01 |
| | Debhata, Satkhira | Low to high stress (very slight saline to slight saline) | 2.47–6.99 |
| | Koyra, Khulna | Medium stress (slight saline to moderate saline) | 7.85–13.20 |
| ALART | BRRI Gazipur | No stress (Favorable) | |
| | Debhata, Satkhira | Low to high stress (very slight saline to moderate saline) | 3.25–7.09 |
| | Assasuni, Satkhira | Low stress (Non saline to very slight saline) | 3.90–6.05 |
| | Batiaghata, Satkhira | Low to medium stress (very slight saline to slight saline) | 2.70–7.35 |
| | Paikgacha, Khulna | High stress (very slight saline to moderate saline) | 4.13–11.58 |
| | Kalapara, Barguna | Low stress (Very slight saline to slight saline) | 3.53–7.01 |
| | Pathorghata, Barguna | Low to medium stress (very slight saline to slight saline) | 3.21–7.83 |
| PVT | Tala, Satkhira | Medium stress (Very slight saline to slight saline) | 4.07–8.17 |
| | Debhata, Satkhira | Low to high stress (very slight saline to moderate saline) | 3.10–12.29 |
| | Kaliganj, Satkhira | High stress (very slight saline to strong saline) | 3.55–16.17 |
| | Dumuria, Khulna | Low stress (Very slight saline to slight saline) | 3.61–6.79 |
| | Paikgacha, Khulna | High stress (Very slight saline to moderate saline) | 4.23–11.79 |
| | Batiaghata, Khulna | Low to medium stress (very slight saline to slight saline) | 3.40–8.50 |
| | Rampal, Bagerhat | Low stress (Very slight saline to slight saline) | 4.08–7.53 |
| | Kalapara, Barguna | Low to medium stress (very slight saline to slight saline) | 3.93–6.99 |

## 2.4 Data analysis

The grain yield data for tested genotypes in three consecutive years in different locations were utilized to execute a combined ANOVA analysis to show the effects of genotype (G), environment (E), and their relationships. The AMMI analysis, heritability for every trial, and standard error of mean were undertaken through STAR (version 2.0.1) and PB Tools (version 1.3; http://bbi.irri.org/products) software. Pearson's correlation coefficients in various locations based on grain yield were assessed utilizing the R (http://www.r-project.org/) chart.Correlation() with the "*Performance Analytics*" package (version 4.0.2). During dry season, a portable electrical conductivity (EC) meter (HANNA, HI 8733) was used to collect weekly data on the water salinity in experimental plots. The different ranges of water salinity were compiled in Fig 2. For constructing the GIS map, we utilized QGIS software version 3.8.2 (https://timdocs.qgis.org/). We have followed the steps below: at first, we extracted the country **shape file** and different the sub-divisions/districts of Bangladesh. After adding all required shape files, then (software command) go to **project> New print composer**, give it a name, then we found an interface to create a map layout, then clicked on **add new map** and draw a box. We checked our study area where we wanted to visualize the study map in the specific box. Then switch to **main properties** and find the **scale** to resize the map in the box. Subsequently, we set up the latitude and longitude in the map then clicked on **grid/add a new grid (+icon)**. After that, we checked the **Draw coordinates** option to visualize the latitude and longitude in the map. To remove the fraction number, put on 0 in the **coordinate precision** option. Then clicked on the **add new label** and drop in the box and write the map title. Finally, we changed the font, font style, and font size from the **Font** option of the **Appearance** menu.

## 2.5 AMMI stability values (ASV) and yield stability index (YSI)

The ASVs were performed to calculate the yield consistency of the various genotypes recommended by [29]. The ASVs were derived from the AMMI, and their estimates were based on

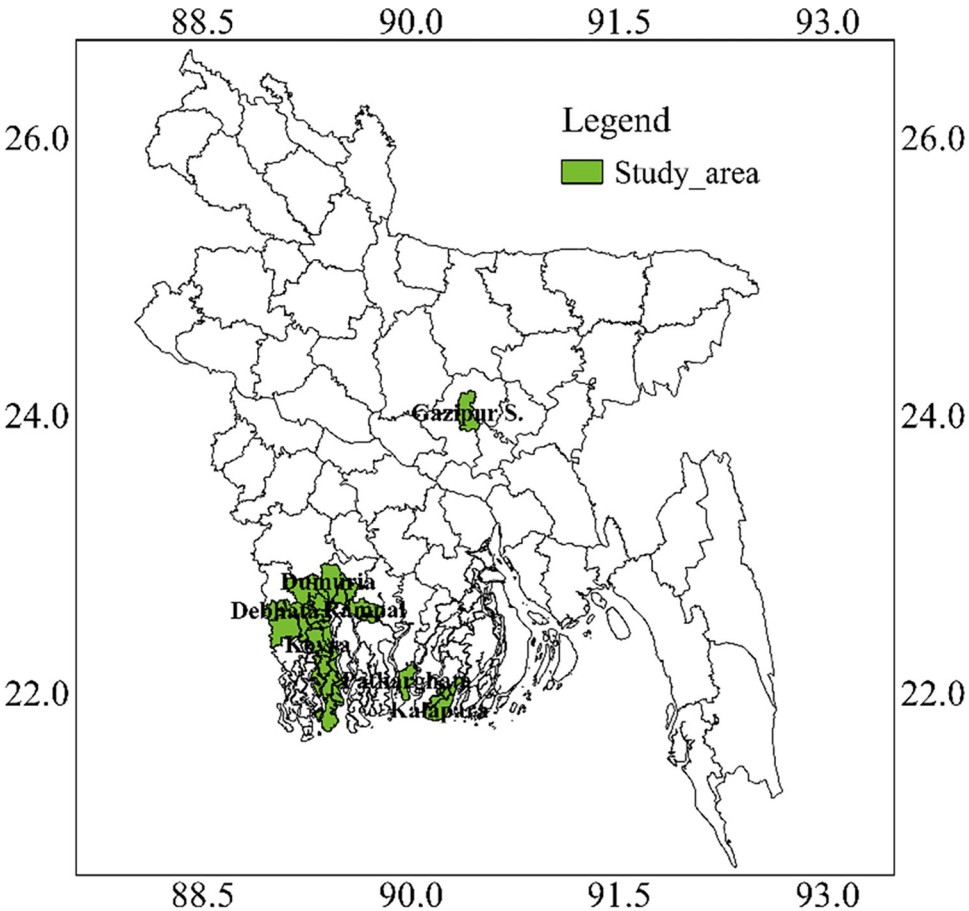

**Fig 2. Variety release system for BRRI developed breeding lines and exotic introduced advanced breeding materials in Bangladesh.**

each individual's surroundings and genotype, which conferred a proportionate role for IPCA1 and IPCA2 using the following formula: ASV = [{(SSIPCA1 ÷ SSIPCA2) (IPCA1 score)}$^2$ + (IPCA2 score)$^2$] $^{1/2}$

Yield stability index (YSI) for individual genotypes which combines both mean grains yield and ASV index was estimated as YSI = RASVi + RYi. Here, the rank of the AMMI stability value for the ith genotype is RASVi, while the rank of the mean grain yield for the ith genotype across environments is RYi [18].

## 2.6 Estimation of breeding value

We determined the estimated breeding value (EBV) using R (version 3.5.1). We calculated the accuracy/reliability of EBV derived from prediction error variance (PEV) as the square root of 1 minus the ratio between PEV and additive genetic variance (VA) i.e., rel < −sqrt(1-PEV/VA).

## 2.7 Grain quality analysis with different parameters

At the Bangladesh Rice Research Institute (www.brri.gov.bd) in Gazipur, Bangladesh, the Grain Quality and Nutrition (GQN) Division laboratory conducted the study of grain quality. Entire and fresh grains were taken for grain quality analysis without broken, discolored, insect,

or disease-infested grain. Dipti et al. [30] have provided essential guidelines for conducting the research.

## Milling properties

A husked rice grinder dehulled 200 grams of unhusked paddy from each sample to demonstrate chalkiness. Therefore, polished rice (10%) for 70 seconds Satake (husked) grain-evaluating mill TM05. One thousand entire grains except for any broken or insect attacks were taken and weighed to measure 1000-grain weight. Milled rice outturn was determined as the percentage of milled rice. Head rice output was determined by explicitly and manually separating broken rice as a percentage of head rice.

## Physical properties

Slide calipers were used to gauge the length and width (in mm) of the milled rice. By dividing the milled rice's length by its breadth, the length-to-breadth ratio was determined. Milled rice was divided into three classes for the purpose of calculating size: long (>6 mm), medium (5–6 mm), and short (<5 mm) [30]. Milled rice was categorized into three types based on shape: slender (more than 3.0), bold (2.0–3.0), and short (less than 2.0) [30].

The presence of a white belly, white center, and intensity of translucence are used to visually evaluate the chalkiness of the kernel. Four classes were usually used to group the existence of white belly or chalkiness of endosperm of milled rice: none (Tr = 0), less than 10% ($Wb_1 < 10\%$), 10% to 20% ($Wb_1 < 10–10\%$), more than 20% ($Wb_1 > 20\%$) [30].

## Chemical properties

For chemical analysis, the polished rice was crushed in an Udy Cyclon sample grinder. Based on [31] recommended Iodine-binding method, the amylose content was calculated. The amylose composition of milled rice was categorized using a five-scale system: Waxy (0–2%), Very low (3–9%), Low (10–19%), Intermediate (20–24%), and High (25%).

Based on the 16.8% nitrogen content of the main rice protein fraction gluten in [32] calculation, nitrogen was used to calculate the protein content and multiplied by a factor of 5.95 [30, 33].

Six whole-milled rice grains were dispersed in triplicate in 10 ml of 1.7% potassium hydroxide (KOH) for 23 hours at normal temperature, and the alkali spreading value (ASV) was computed and evaluated in accordance with [34]. Additionally, there are three classifications of alkali spreading value (ASV): High (1.0–3.0), Intermediate (4.0–5.0), and Low (6.0–7.0).

## Cooking properties

When the cooking time was assessed, 90% of the cooked rice had gelatinized. The ratio of cooked rice length to uncooked rice length, as determined by a slide caliper, is known as the elongation ratio. The imbibition ratio, which is determined using the water displacement technique, is the increase in cooked rice volume over uncooked rice [30]. After five grams of milled rice and 50 ml of water were added to a graduated cylinder, the volume difference was recorded. 5 g of milled rice was first cooked in order to determine the amount of cooked rice. The volume difference was then calculated after adding the cooked rice to the cylinder.

### 2.8 Trait identification by SNP genotyping

The proposed lines were genotyped with 20 gene-based single nucleotide polymorphism (SNP) markers to identify desired QTLs progressed through the International Rice Research

Institute (IRRI; https://www.irri.org/ (accessed on 14 January 2023); https://gsl.irri.org/ (accessed on 20 January 2023); [35] employing Kompetitive allele-specific PCR (KASP) assay for high-throughput bi-allelic categories of SNP with Intertek (https://www.intertek.com/ agriculture/agritech/ (accessed on 28 January 2023) as an outsourcing provider. The SNP markers connected with the trait of benefits such as snpOS00038, snpOS00445 refers amylose content; snpOS00024 refers chalkiness; snpOS00396 refers grain number; snpOS00397, snpOS00398, snpOS00409, snpOS00410, and snpOS00411 refers salt tolerance; snpOS00459 refers anaerobic germination; snpOS00403 refers cold tolerance; snpOS00006, snpOS00478, snpOS00468, snpOS00451 refers rice blast; snpOS00054 (AG), snpOS00493, snpOS00061 refers bacterial leaf blight; snpOS00430, snpOS00442, and snpOS00486 refers brown planthopper; and snpOS00466, snpOS00467 for gall midge were assessed. These 20 SNP are associated with extremely important traits. These trait-connected SNP are utilized to recognize the basis of the suitable genotype on the accessibility of well-known traits.

## 3. Results

### 3.1 Regional yield trial (RYT) for measuring the regional adaptation and suitability

**3.1.1 Geographic location and salinity levels.** There are four experimental sites for testing the regional suitability and adaptation in the RYT trial. All locations showed a representative performance. The useful information on the experimental sites is given in Table 2.

The salinity level depends on the environment, temperature, and precipitation which was measured at seven days intervals. This was done from the planting date to the tested genotype's flowering stage. Kaliganj found an extremely high level of salinity where most of the genotypes died and did not find the yield. The salinity levels of Kaliganj of the studied plot range from 3.96 dS/m to 15.01 dS/m while, 2.47 dS/m to 6.99 dS/m was detected in Debhata, which has a lower salinity level (Fig 3). All three locations had a significantly positive correlation except Kaliganj, which has no yield data due to extreme salinity. A highly significant positive correlation (0.69) showed between Koyra and Debhata nevertheless, the significant correlation between Assasuni and Debhata (0.36); Koyra and Assasuni (0.52) are also positive but not higher (S1 Fig) representing that, both Debhata and Assasuni had comparatively lower salinity level and Koyra (higher salinity) is dissimilar from Assasuni and Debhata.

The AMMI analysis of variance for a combined mean yield of fifteen rice genotypes from three locations exhibited that the greater part of the total sum squares elucidated by genotypic effects (50.89%) followed by Location × Genotype interaction (LGI) (24.75%) and locations effects (9.74%) (Table 2). The ANOVA of AMMI explained significant variation among 15 rice genotypes and three locations. This shows that genotype, locations, and their interactions with genotypes are all indicators of the yield of studied genotypes. AMMI analysis additionally subdivided the LGI into the first two multiplicative principal components explicitly PC1 and PC2 with involvement of 55.1% and 44.9% of LGI sum of squares (Table 3).

**Table 2. Experimental sites and their geographical position of the current study.**

| Upazila | District | Longitude E | Latitude N | CV (%) | Yield (tha$^{-1}$) | IPCA1 | IPCA2 |
|---------|----------|-------------|------------|--------|------------------|-------|-------|
| Assasuni (E1) | Satkhira | 89˚08'58.25" | 22˚34'37.51" | 9.08 | 5.77 | -0.96 | 0.46 |
| Debhata (E2) | Satkhira | 88˚28'21.32" | 22˚64'32.17" | 14.41 | 5.29 | 0.90 | 0.56 |
| Koyra (E3) | Khulna | 89˚19'46.80" | 22˚27'13.70" | 15.54 | 5.26 | 0.06 | -1.02 |
| Kaliganj (E4) | Satkhira | 89˚09'27.21" | 22˚40'22.71" | - | - | - | - |

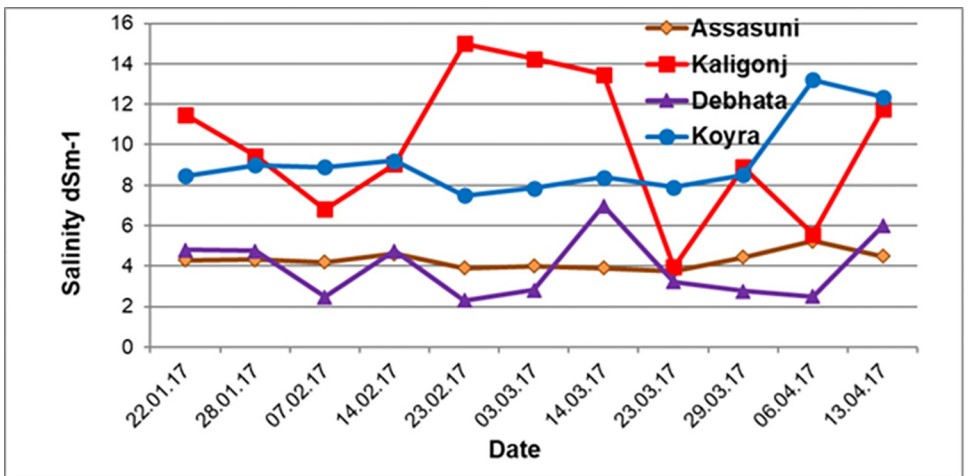

**Fig 3. Salinity condition of the different experimental sites during *Boro* 2016–17 in Regional Yield Trial (RYT).** In this figure, the highest salinity peak (15 ds/m$^2$) appears in the Kaligang location followed by Koyra, the lowest peak observed in Assasuni. The highest salinity data were recorded on 23 February, when plants continue the vegetative growth phase.

**3.1.2 Evaluation of tested genotype among locations.** Grain yield analysis of elite rice showed that the average yield mean is 5.51 tha$^{-1}$. Ten genotypes revealed an above-average yield in Assasuni. Seven genotypes in Debhata and 8 genotypes in the Koyra locations outclassed the average yield of the consistent genotypes in a specific location (Table 4). The average plant height varied from 77 cm to 100 cm. The time required to reach maturity ranged from 142 to 155 days. In Assasuni, the range of grain yield was 4.82–6.53 tha$^{-1}$, with an average of 5.82 tha$^{-1}$; in Debhata, it was 3.74–6.37 tha$^{-1}$, with an average of 5.35 tha$^{-1}$; and in Koyra, it was 3.59–6.33 tha$^{-1}$, with an average of 5.35 tha$^{-1}$ (Table 4).

Genotypes are selected based on AMMI stability value (ASV) which is essential due to the furthermost stable genotypes may not continually yield the highest yields and otherwise display steady performance and usual adaptability that genotypes were nominated based on ASV. The most stable genotypes were those with lower ASV and IPCA values. Smaller ASV scores and a higher mean yield are characteristics of an optimal stable genotype. Therefore, G6, G7, and G9 exhibited the lower ASVs (0.15, 0.14, and 0.16) respectively, and medium grain yield

**Table 3. Analysis of AMMI variance for grain yield (tha$^{-1}$) of fifteen rice genotypes in RYT among three different locations in Bangladesh.**

| Source of variation | DF | Sum of Squares | Mean Squares | F Value | Pr (>F) | Variability explained | |
|---|---|---|---|---|---|---|---|
| | | | | | | % GL | % TSS |
| Location (L) | 2 | 4.45 | 2.23* | 6.94 | 0.0750 | - | 9.74 |
| Rep within location | 3 | 0.96 | 0.32* | 2.19 | 0.1044 | - | 2.11 |
| Genotype (G) | 13 | 23.24 | 1.78** | 12.21 | 0.0000 | - | 50.89 |
| Location: Genotype | 26 | 11.30 | 0.43** | 2.97 | 0.0010 | - | 24.75 |
| IPCA1 | 14 | 4.67 | 0.33 | 1.80 | 0.0000 | 55.1 | - |
| IPCA2 | 12 | 3.81 | 0.31 | 1.71 | 0.0000 | 44.9 | - |
| IPCA3 | 10 | 0.00 | 0.00 | 0.00 | 1.00 | 0.00 | - |
| Pooled Error | 39 | 5.71 | 0.15 | - | - | - | 12.51 |
| Total | 83 | 45.67 | | - | - | - | - |

*and ** significant at 5% and 1% level respectively, % TSS = Percentage total sum of square; % GL = Percentage (Genotype × Location).

Table 4. Yield, agronomic characters, and estimated breeding values of the proposed salinity tolerant rice during regional yield trial (RYT). Boro 2016–2017.

| Geno code | Designation | Parentage | Days to maturity (days) | Plant height (cm) | Assasuni (E1) yield (tha⁻¹) | Debhata (E2) yield (tha⁻¹) | Koyra (E3) yield (tha⁻¹) | Mean yield (tha⁻¹) | IPCA1 | IPCA2 | ASV | EBV (tha⁻¹) | Rel | Yield Rank (RYi) | ASV Rank (RASVi) | YSI |
|---|---|---|---|---|---|---|---|---|---|---|---|---|---|---|---|---|
| G1 | BR8940-B-17-4-7 | IR72593-B-3-2-2-2/ BRRI dhan47 | 143±0.56 | 85±4.23 | 6.15ab | 5.05a-e | 4.89bc | 5.36 ±0.34 | -0.36 | 0.27 | 0.50 | -0.12 | 0.70 | 9 | 6 | 15 |
| G2 | BR8943-B-20-9-22 | BRRI dhan47/ IR69337-AC2-2-2 | 144±0.67 | 82±5.60 | 5.73ab | 4.50de | 4.40bc | 4.88 ±0.34 | -0.43 | 0.26 | 0.57 | -0.50 | 0.57 | 14 | 5 | 19 |
| G3 | IR86385-85-2-1-B | IRRI 149/ IR61920-3B-22-2-1 | 145±0.75 | 100 ±5.53 | 5.66ab | 3.74e | 5.57ab | 4.99 ±0.40 | -0.69 | -0.78 | 1.13 | -0.42 | 0.79 | 13 | 1 | 14 |
| G4 | IR83484-3-B-7-1-1-1 | IRRI 113/BR 40 | 146±0.82 | 84±4.39 | 5.61ab | 6.37a | 6.33a | 6.10 ±0.18 | 0.69 | -0.29 | 0.87 | 0.47 | 0.79 | 3 | 2 | 5 |
| G5 | IR87872-7-1-1-2-1-B | AT 401/ IR73571-3B-14-1 | 144±1.02 | 83±1.26 | 5.36ab | 5.25a-d | 4.67bc | 5.09 ±0.16 | 0.17 | 0.28 | 0.34 | -0.34 | 0.59 | 10 | 7 | 17 |
| G6 | IR86385-117-1-1-B | IRRI 149/ IR61920-3B-22-2-1 | 147±1.01 | 82±2.67 | 5.50ab | 4.87b-e | 4.78bc | 5.05 ±0.24 | -0.09 | 0.10 | 0.15 | -0.37 | 0.79 | 12 | 14 | 26 |
| G7 | IR87870-6-1-1-1-B | AT 401/CSR 2 | 150±1.30 | 92±2.99 | 6.37a | 5.69a-d | 5.73ab | 5.93 ±0.16 | -0.11 | 0.03 | 0.14 | 0.34 | 0.79 | 5 | 15 | 20 |
| G8 | BR8980-4-6-5 | BRRI dhan45/ BRRI dhan47 | 140±1.15 | 81±3.07 | 5.54ab | 4.88b-e | 4.80bc | 5.07 ±0.17 | -0.11 | 0.10 | 0.16 | -0.35 | 0.55 | 11 | 13 | 24 |
| G9 | BR8980-B-1-3-5 | BRRI dhan45/ BRRI dhan47 | 142±1.22 | 77±4.28 | 5.60ab | 5.56a-d | 5.51ab | 5.56 ±0.03 | 0.24 | -0.08 | 0.30 | 0.04 | 0.57 | 8 | 8 | 16 |
| G10 | BR8992-B-18-2-26 | BRRI dhan47/ FL478 | 145±1.23 | 79±4.45 | 5.85ab | 5.77a-d | 5.73ab | 5.78 ±0.14 | 0.21 | -0.07 | 0.26 | 0.22 | 0.79 | 7 | 10 | 17 |
| G11 | HHZ5-DT20-DT2-DT1 | Huang-Hua-Zhan/ OM1723 | 155±1.50 | 91±1.09 | 5.80ab | 6.18ab | 6.47a | 6.15 ±0.14 | 0.49 | -0.40 | 0.71 | 0.51 | 0.79 | 2 | 3 | 5 |
| G12 | HHZ12-SAL2-Y3-Y2 | Huang-Hua-Zhan/ Teqing | 155±1.53 | 96±1.01 | 6.53a | 6.12ab | 6.50a | 6.38 ±0.20 | 0.06 | -0.25 | 0.26 | 0.70 | 0.79 | 1 | 11 | 12 |
| G13 | BRRI dhan28 (S. Ck) | BR 6/PURBACHI | 142±0.86 | 87±4.57 | 4.82b | 4.73c-e | 3.59c | 4.38 ±0.26 | 0.14 | 0.63 | 0.65 | -0.90 | 0.79 | 15 | 4 | 19 |
| G14 | BRRI dhan67 (Ck) | IR 61247-3B-8-2-1/ BR 36 | 144±1.02 | 94±6.94 | 6.33a | 5.42a-d | 5.63ab | 5.80 ±0.17 | -0.23 | -0.02 | 0.27 | 0.23 | 0.79 | 6 | 9 | 15 |
| G15 | Binadhan-10 (Ck) | R 42598-B-B-B-B-12/ NONA BOKRA | 147±1.56 | 98±5.59 | 6.45a | 6.05a-c | 5.65ab | 6.05 ±0.14 | 0.02 | 0.24 | 0.24 | 0.43 | 0.71 | 4 | 12 | 16 |
| | Mean | | 145 | 87 | 5.82 | 5.35 | 5.35 | 5.51 | | | | | | | | |
| | LSD (0.05) | | 3.03 | 9.43 | 0.79 | 0.74 | 0.72 | 0.74 | | | | | | | | |
| | H²b | | 0.91 | 0.77 | 0.59 | 0.86 | 0.89 | 0.80 | | | | | | | | |

Here, ASV = AMMI stability value; EBV = estimated breeding values; Rel = Reliability; H²b = Heritability; RASVi = Rank of the AMMI stability value for the ith genotype; RYi = Rank of the mean grain yield for the ith genotype across environments; Yield stability index (YSI)

(mean yield: 5.05±0.24, 5.93±0.16, and 5.07±0.17 tha$^{-1}$), respectively (Table 4). Additionally, G12 was the high-yielder genotype (6.38 tha$^{-1}$) with comparatively low ASV (0.26). These reports showed that those genotypes are more stable compared to other genotypes; including G4, G7, G11, G14, and G15 were the top five high yielder genotypes (6.10±0.18, 5.93±0.16, 6.15±0.14, 5.80±0.17, 6.05±0.14 tha$^{-1}$, respectively), then had higher ASV (0.87, 0.14, 0.71, 0.27, and 0.24 respectively) were recognized as promising genotypes for the *Boro* or dry season. The lower value of yield stability index (YSI:5) is considered as the most stable with a high grain yield. In this case, G4 (IR83484-3-B-7-1-1-1) and G11 (HHZ5-DT20-DT2-DT1) were the most stable genotypes (Table 4). For the correlations of agronomic traits, yield (t/ha) and days to maturity (days) showed a positive significant association with plant height (cm) and panicle length (cm) Besides, unfilled grain revealed negatively correlated with yield and all others traits (S4 Fig)

The estimated breeding values (EBV) for entire genotypes wide-ranging from—0.12 to 0.70 tha$^{-1}$. The higher EBV was found in G12: HHZ12-SAL2-Y3-Y2 (0.70 tha$^{-1}$) and the lower one was for G1: BR8940-B-17-4-7 (- 0.12 tha$^{-1}$) along with a reliability of 79% (Table 4). The eight genotypes were between 0.04 to 0.70 EBVs and between—0.12 to 0.90 for seven genotypes. The genotypic correlations among tested genotypes and the heritability of yield were also assessed. Reliability is a vital indicator for calculating the precision of estimated breeding values (EBV) of individual genotypes [36]. Reliability is the square of accuracy ($r^2$), where accuracy (r) is the correlation parameters among EBV and true breeding value (TBV). The reliability is 55–79% indicating a moderate level of accuracy.

**3.1.3 Participatory varietal selection (PVS) approach for selecting the best one.** Using the PVS method, farmers and other participants (scientists, representatives of government and non-government organizations, and local leaders) evaluated the acceptability of promising rice genotypes and ranked them [37]. A group of twenty male farmers and ten female farmers participated in this approach. At maturity, this work was completed at 80% (The majority of contemporary varieties matured). Each farmer was provided with four paper ballots (two with tick marks for chosen entries and two with cross marks for worse entries) and they were given necessary instruction on how to cast two ballots for two best lines and the other two ballots for worst lines. After voting, the experts tabulated the results and used a flip chart board to present them to the farmers (S1 Table).

Farmers' preference rankings for the tested genotypes placed IR83484-3-B-7-1-1-1 (PVS-4) first at Debhata and Kaliganj, HHZ5-DT20-DT2-DT1 (PVS-11) at Koyra, and HHZ12-SAL2-Y3-Y2 (PVS-12) at Assasuni. Besides, BRRI dhan67 (Ck) ranked the second choice in PVS at Debhata, Binadhan-10 (Ck) in Assasuni, HHZ5-DT20-DT2-DT1 in Kaliganj, and IR83484-3-B-7-1-1-1 in Koyra by both male and female farmers. The preferred and non-preferred (worst) genotypes including farmers reaction/feedback are shown in Table 5.

**3.1.4 Best genotype identification in each location.** The diagram with a polygon preview of the GGE biplot is a suitable technique to identify the winning genotypes and their suitable environments. A polygon is formed by connecting all the genotypes that are far from the origin of the biplots. The biplot is partitioned into numerous sectors by a perpendicular line that appears from the biplot's origin and extends outside of the polygon (Fig 4). The genotypes found at each sector's vertices had the best performance among genotypes in that sector in different situations. Gen3, Gen4, Gen5, Gen9, and Gen10 are the vertex genotypes in this study. Debhata (E2) is situated in the sector in which Gen10 was the vertex cultivar. This indicates that Gen10 was the prime cultivar for grain yield in the Debhata location. The Gen4 was better one in the location Assasuni and Koyra, where Gen4 was the winning genotype. Gen5 and Gen9 as the vertices of no habitats fell into sectors, indicating that these cultivars were unsuitable for any of the environments. The environments fall into the same vector whereas the

**Table 5. Farmers' feedback on best/ranked one genotype and worst genotypes in various locations of salt-stress prone environments under participatory variety selection.**

| Comments on the overall performance of the ranked one genotype | |
| --- | --- |
| Assasuni (PVS-12) | The plant is medium in height, has a large number of tillers, fewer empty spikelet's, a lengthy panicle height, and produces more yield. |
| Debhata (PVS-4) | High yielder, plant height is short, early flowering, Less disease, and pest infestation, longer panicle size, high grain weight. |
| Koyra (PVS-11) | Slender grain, small plant size, many tillers, lengthy panicle, and disease-free. |
| Kaliganj (PVS-4) | Low unfilled grain, excellent plant type, high tiller density per hill, high salt tolerance, and low disease infestation. |
| **Comments on the overall performance of the worst genotype** | |
| Assasuni (PVS-3) | The degree of disease and insect infestation, the number of empty spikelet's, the number of tillers and panicles, and the size of the grain in each panicle. |
| Debhata (PVS-10) | Late maturity, smaller, shorter, and fewer tillers; disease and bug infestation. |
| Koyra (PVS-2) | Bold grain, very short plant, fewer tillers, and late maturity. |
| Kaliganj (PVS-1) | Very sensitive to salinity, has a high number of unfilled grains, appearance of plant type is not good. |

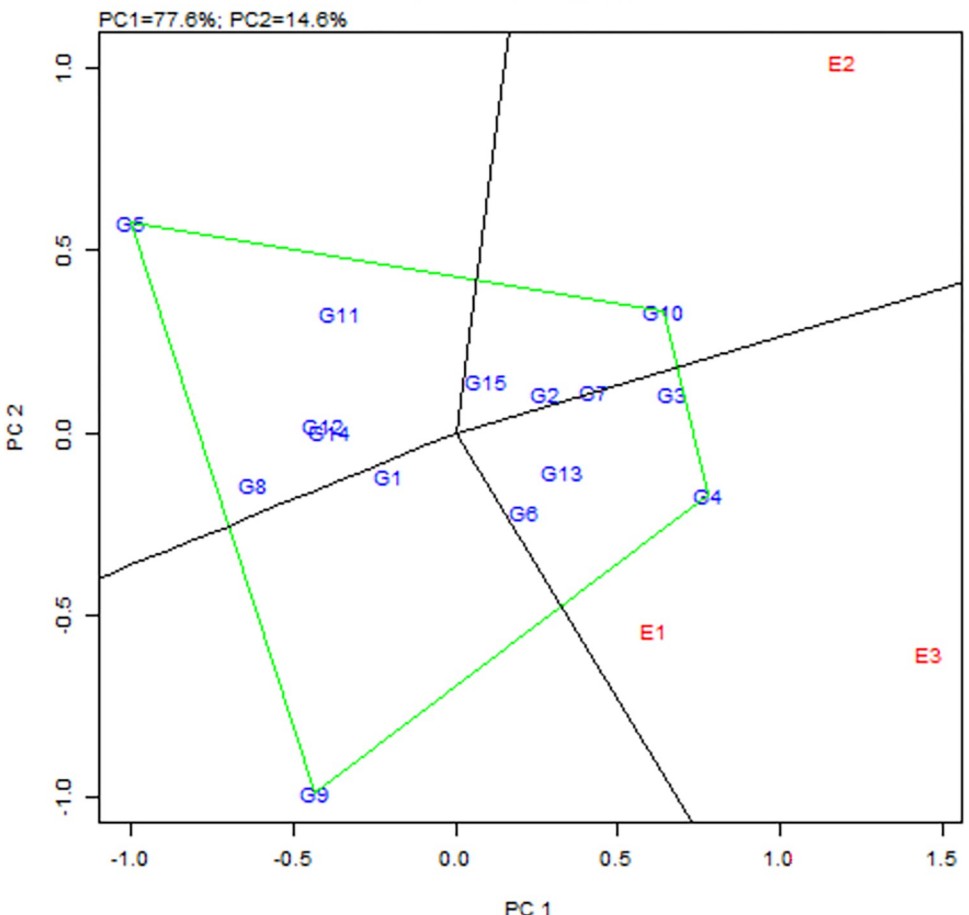

**Fig 4. The polygon preview of the GGE biplot for the identification winning as well as best rice genotypes with associated environments.** Here, G4 and G10 are the winning genotypes in the Assasuni(E1), Koyra(E3), and Debhata (E2), respectively.

Table 6. Combined analysis of variance of tested genotypes in different seven locations.

| Source | DF | Sum of square | Mean square | F value | Pr (>F) | %TSS |
|---|---|---|---|---|---|---|
| Location | 6 | 27.18 | 4.53 | 173.79 | 0.0000 | 61.96 |
| Rep within location | 14 | 0.36 | 0.02 | 2.26 | 0.0162 | 0.82 |
| Genotype | 4 | 2.23 | 0.55 | 48.45 | 0.0000 | 5.08 |
| Location: Genotype | 24 | 13.42 | 0.55 | 48.41 | 0.0000 | 30.59 |
| Error variance | 56 | 0.64 | 0.02 | | | 1.46 |
| Total | 104 | 43.86 | | | | |

DF = Degree of freedom, %TSS = Percentage Total Sum of Square, Pr (P-value corresponding to the F value of a specific effect) or significance at the 0.01 level of probability

environment of Debhata falls into the other vector which means two locations, Assasuni and Koyra, might possess similar climatic conditions.

## 3.2 Advanced lines adaptive research trial (ALART) for testing adaptability in the farmer's field under salt-affected areas

At farmers' fields in various agroecological zones, the ALART was experimented with to assess the yield performance and adaptation of the selected advanced lines. In this study, the selected genotypes like IR83484-3-B-7-1-1-1, HHZ12-SAL2-Y3-Y2, and HHZ5-DT20-DT2-DT1 were assessed in the different salt-affected areas especially the southern part of Bangladesh. Highly significant main effects (P <0.01) of genotypes, environments, and GE interaction were found in the combined analysis of variance for grain yield of rice genotypes across different locations (Table 6). A total of 61.96% of the variation was caused by location whereas, the genotypes contributed only 5.08% of the total variation. The GL interaction main effect made up 30.59% of the variation, and error variance made up only 1.46%.

The breeding line IR83484-3-B-7-1-1-1 was chosen, due to its medium bold grain and high salinity tolerance ability, which is popular for greater Barishal regions. The plant height of this line is 103 cm and the average growth duration was 150 days.

The HHZ12-SAL2-Y3-Y2 (6.34 tha$^{-1}$) and HHZ5-DT20-DT2-DT1(6.36 tha$^{-1}$) produce statistically similar yields compared with the tolerant checks BRRI dhan67 (6.41 tha$^{-1}$) but greater than the susceptible checks BRRI dhan28 (6.01 tha$^{-1}$) (Table 7). The salinity tolerance capacity of these genotypes is more than the check varieties. The plant height of the HHZ12-SAL2-Y3-Y2 is 99 cm, whereas HHZ5-DT20-DT2-DT1 is 94 cm. The average growth duration of these genotypes is 147 and 148 days, respectively. Based on high salinity tolerance ability, higher yield, good grain quality, and farmer's acceptability, the above genotypes are further proceeding for the proposed variety trial for checking the suitability and adaptability of the salinity-prone areas.

## 3.3 Proposed variety trial (PVT) for variety release in commercial cultivation

The NSB team evaluated the proposed genotype on-farm for the PVT before recommending its release as a new variety. There are eight locations that consist of three severe salinity-affected areas. In Kaliganj, Debhata and Paikgacha initiate a tremendously higher level of water salinity wherever some of the tested genotypes have complexly damaged which is very susceptible to salt water and can't withstand salt stress and ultimately did not give the yield. Some genotypes have shown their survivability with a minimum reduction of yield indicating

**Table 7. Salinity-tolerant genotypes' performance in terms of yield in various hotspot regions during the growing seasons of 2017–2018.**

| Designation | Plant height* (cm) | Growth duration* (days) | Yield (tha$^{-1}$) in different locations | | | | | | | Mean |
| --- | --- | --- | --- | --- | --- | --- | --- | --- | --- | --- |
| | | | S1 | S2 | S3 | S4 | S5 | S6 | S7 | |
| IR83484-3-B-7-1-1-1 | 103 | 150 | 6.45c | 6.42d | 6.22d | 6.16c | 6.18b | 6.12a | 6.15d | 6.24 ±0.03 |
| HHZ12-SAL2-Y3-Y2 | 99 | 147 | 7.12a | 6.97a | 7.24a | 4.80d | 5.89c | 5.66c | 6.72b | 6.34 ±0.19 |
| HHZ5-DT20-DT2-DT1 | 94 | 148 | 7.04a | 6.71b | 7.13a | 4.69d | 6.53a | 5.91b | 6.51c | 6.36± 0.17 |
| BRRI dhan28 (Sus. Ck.) | 97 | 140 | 6.61c | 6.24d | 6.46c | 5.97b | 5.08d | 5.35d | 6.30d | 6.01± 0.12 |
| BRRI dhan67 (Std & tol. Ck.) | 105 | 143 | 6.85b | 6.65b | 6.91b | 5.59c | 6.14b | 5.74b | 7.02a | 6.41± 0.12 |
| H2 | | | 0.79 | 0.90 | 0.96 | 0.93 | 0.90 | 0.93 | 0.99 | |

Mean of seven locations (S1 = BRRI Gazipur, S2 = Debhata, S3 = Assasuni, S4 = Batiaghata, S5 = Paikgacha,

S6 = Kalapara, S7 = Pathorghata)

* indicates average value. There is no noticeable difference between means with the same letter.

their tolerance to salinity. The salinity levels of Kaliganj of the experimental sites range from 3.55 dS/m to 16.17 dS/m; 3.1 dS/m to 12.29 dS/m in Debhata while 4.23 dS/m to 11.79 dS/m was recorded in Paikgacha (S2 Fig).

On the other hand, Tala, Dumuria, and Rampal appear moderate salinity range (2.55–8.17 dS/m). It is well popular information that low to moderate level of salinity is helpful for better growth and development of plants by enhancing their hybrid vigor. The experimental site Dumuria revealed a highly significant positive correlation among Batiaghata (0.83) and Rampal (0.79), however, the significant correlation among Kalapara and Rampal (0.80); Batiaghata and Rampal (0.67) are likewise significantly positive correlation (Fig 5) performing that both those locations are relatively low saline, although Kalapara is different from Batiaghata and Rampal, even so far away districts.

The genotypes IR83484-3-B-7-1-1-1 and HHZ5-DT20-DT2-DT1 can tolerate 14 dS/m salinity at the seedling stage. Additionally, it can produce grain yield with 8–10 dS/m salinity level through all the salt-sensitive stages from vegetative to reproductive stages. Both of them can tolerate more salinity compare to BRRI dhan67. The average yield potential of IR83484-3-B-7-1-1-1 and HHZ5-DT20-DT2-DT1 are 4.90±0.24 tha$^{-1}$ and 5.45±0.32 tha$^{-1}$ even though they can produce 3.93 to 5.95 tha$^{-1}$ and 4.14 to 6.58 tha$^{-1}$ depending on the salinity level, respectively (Table 8). Both genotypes have the potential to produce more than 7.0 tha$^{-1}$ in a favorable environment with proper management. The susceptible check BRRI dhan28 and tolerant check BRRI dhan67 were completely damaged in the severely affected salinity areas but in the low to medium areas, BRRI dhan28 gives considerable grain yield (2.54 tha$^{-1}$) in Tala, 1.74 tha$^{-1}$ in Dumuria, 0.95 tha$^{-1}$ in Rampal. On the other hand, tolerant check BRRI dhan67 produces a low yield (3.17 tha$^{-1}$) in Rampal compared to other locations. The crop conditions at the maximum tillering stage of the Kaliganj experimental sites are shown in Fig 6.

## 3.4 Estimation of grain quality properties of proposed lines

It is commonly known that non-sticky rice and foods with a high amylose content (%) are favorites of Bangladeshis. As a result, IR83484-3-B-7-1-1-1 and HHZ5-DT20-DT2-DT1 have been developed into a salinity tolerance variety with a high yield and acceptable grain

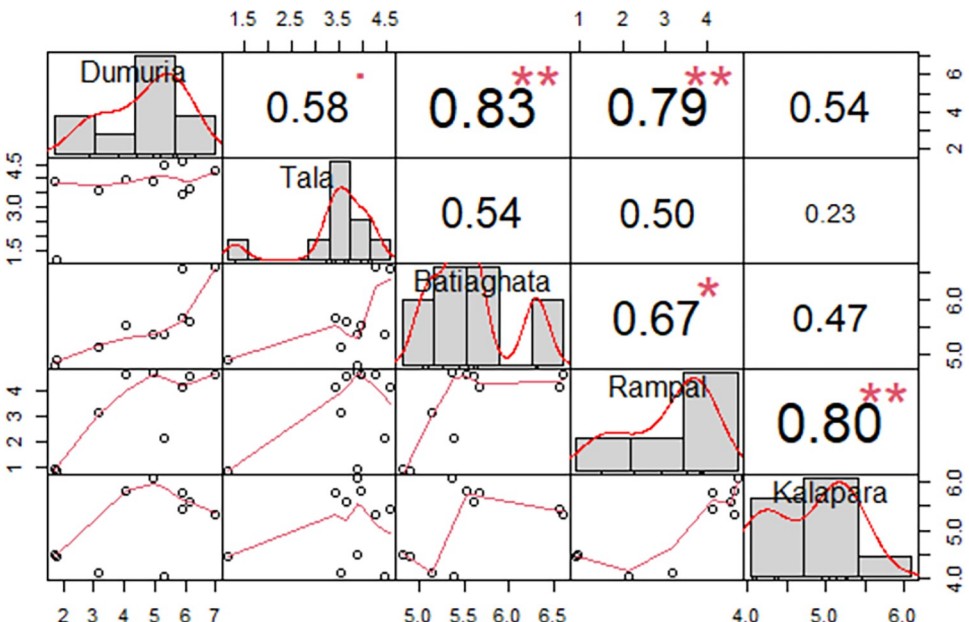

**Fig 5. Correlation among studied locations for yield performance to show their interrelations and effect on tested genotypes.** The correlation coefficient and the level of significance are displayed as stars at the top of the diagonal. * $p \leq 0.05$ and ** $p \leq 0.01$ show significance level.

characteristics. The different physicochemical parameters such as head rice yield (%), Milling outturn (%), Milled rice length and Breath (mm), L/B ratio, size and shape, chalkiness, and cooking properties viz amylose (%), elongation ratio (ER), and imbibition ratio (IR) were shown in Table 9. In order to ensure a high market price, acceptable grain qualities like amylose content must be greater than 25%, total milling outturn must be greater than 66%, and head rice must be greater than 50%. Additional traits include long slender grain (LS), high elongation ratio (ER), and non-chalk or translucence (Tr). The attitude of the flag leaf blade of IR83484-3-B-7-1-1-1 and HHZ5-DT20-DT2-DT1 are semi-erect, widely long, and dark green in color. They have well-exerted panicles with non-shattering behavior in the panicle. The

**Table 8. Estimating the yield of proposed genotypes in all locations, including those with extreme salinity, during the growing season of 2018–19.**

| Genotypes | Yield (tha⁻¹) | | | | | | | | |
|---|---|---|---|---|---|---|---|---|---|
| | S1 | S2* | S3* | S4 | S5* | S6 | S7 | S8 | Mean |
| IR83484-3-B-7-1-1-1 | 3.93b | 1.09 | 2.38 | 4.49a | 1.01 | 5.45c | 4.71a | 5.95a | 4.90±0.24 |
| HHZ12-SAL2-Y3-Y2 | 3.56b | 0.98 | 0.82 | 6.01a | 0.44 | 5.64b | 4.56a | 5.69ab | 5.05±0.31 |
| HHZ5-DT20-DT2-DT1 | 4.44a | 1.04 | 0.83 | 6.44a | 0.21 | 6.58a | 4.14a | 5.40b | 5.45±0.32 |
| BRRI dhan28 (Sus. Ck.) | 2.54c | 0.04 | 0.00 | 1.74b | 0.00 | 4.86e | 0.95c | 4.07d | 2.83±0.54 |
| BRRI dhan67 (tol. Ck.) | 4.01a | 0.73 | 0.00 | 4.23a | 0.00 | 5.14d | 3.17b | 4.47c | 4.20±0.33 |
| Mean ± standard error | 3.69 ±0.30 | - | - | 4.58 ±0.58 | - | 5.56 ±0.19 | 3.40 ±0.48 | 5.12 ±0.24 | 4.47 |
| LSD (0.05) | 0.96 | - | - | 0.82 | - | 0.05 | 0.75 | 0.22 | |
| CV (%) | 26.03 | - | - | 18.03 | - | 10.91 | 11.64 | 15.00 | |

Average of three replications (S1 = Tala, S2 = Debhata, S3 = Kaliganj, S4 = Dumuria, S5 = Paikgacha, S6 = Batiaghata, S7 = Rampal, S8 = Kalapara), No discernible difference exists between means with the same letter.

* Denote genotypic average yield values are excluded from the total mean calculation.

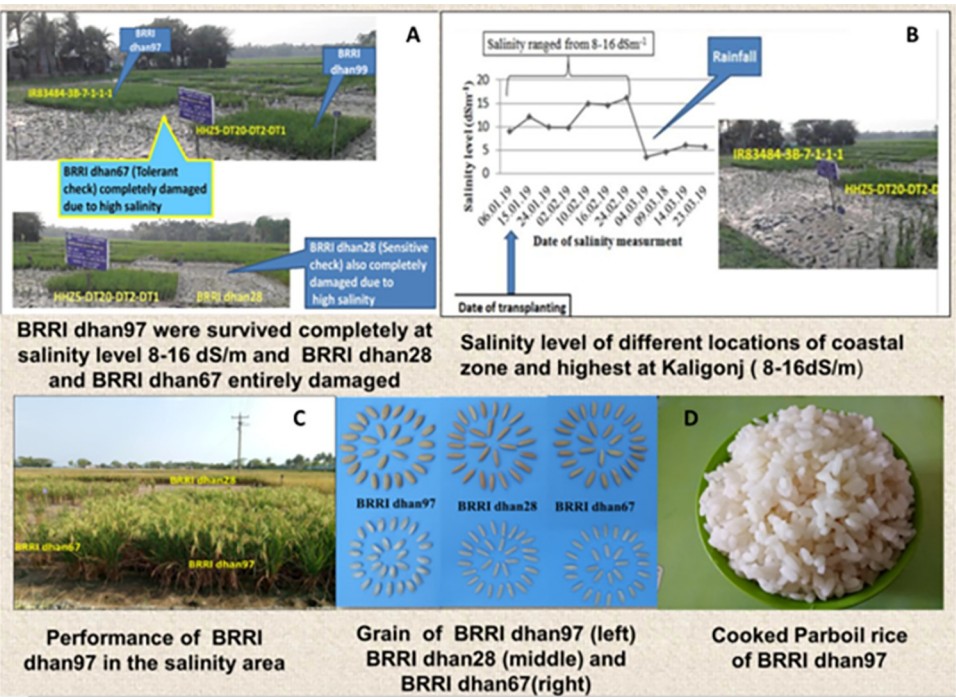

**Fig 6. The field condition at Kaliganj location under salt-stress and grain quality parameters.** The proposed varieties A) IR83484-3-B-7-1-1-1 (BRRI dhan97) and HHZ5-DT20-DT2-DT1 (BRRI dhan99) were completely survived at B) 8–16 dS/m salinity level, whereas the susceptible check BRRI dhan28 and the salt-tolerant check BRRI dhan67 were entirely damaged due to severe salinity stress in the Kaliganj, Satkhira site during *Boro* 2018–2019, C) Performance of BRRI dhan97 at maturity under salt stress, D) grain and cooked rice of BRRI dhan97.

anthocyanin color in the base of the leaf sheath is present in IR83484-3-B-7-1-1-1. Without dehulling, the decorticated grain length was medium and the shape was medium bold in IR83484-3-B-7-1-1-1 and very long in HHZ5-DT20-DT2-DT1. The parboiled and unparboiled milled rice is medium bold in IR83484-3-B-7-1-1-1 and long-slender in HHZ5-DT20-DT2-DT1 in size and shape, translucent, and cooked rice is fluffy (Fig 7). The average thousand-grain weight is 25.5 grams and 22.8 grams; amylose content is 25.2% and 27.1%; protein content is 8.6% and 7.9%; elongation ratio is 1.9 and 1.5; imbibition ratio is 2.7 and 3.1; alkali spreading value is 7.0 and 7.0 of IR83484-3-B-7-1-1-1 and HHZ5-DT20-DT2-DT1, respectively (Table 10).

### 3.5 Trait categorization of proposed lines utilizing trait-based SNP

Two proposed lines with three checks were genotyped and counted 20 trait-based SNP markers to the desired traits by means of grain quality [Wx-A_group {snpOS00445 (C)}, Wx-GBSS-ex10 {snpOS00038 (T)} for amylose content; chalk5_576 {snpOS00024 (G)} for chalkiness; Gn1a_1 {snpOS00396 (T)} for grain number; snpOS00397 (T), snpOS00398 (T), qSES1-2_2: snpOS00409 (C), qSES1- 2_3: snpOS00410 (A), qSES1-2_4: snpOS00411 (T) refers salt tolerance;

qAG3_1: snpOS00459 (C) refers anaerobic germination; qSCT1_1: snpOS00403 (A) refers cold tolerance; Pi-ta: snpOS00006 (C), Pb1: snpOS00478 (T), Pi33_1: snpOS00468 (T), Pi9_1: snpOS00451 (C) for rice blast; xa5-S1_SKEP: snpOS00054 (AG), xa13_1: snpOS00493 (C), Xa21_SKEP: snpOS00061 (C) for bacterial leaf blight; Bph17_3: snpOS00430 (G), BPH32: snpOS00442 (G), BPH9: snpOS00486 (A) for brown plant hopper; and Gm4_3: snpOS00466

**Table 9. List of the gene-based SNP markers with favorable allele used to characterize two proposed lines with three checks for the various traits of interest.**

| Trait | Gene/ QTL | Gene-based SNP with favor able allele | Genotypes | | | | |
|---|---|---|---|---|---|---|---|
| | | | IR83484-3-B-7-1-1-1 | HHZ5-DT20-DT2-DT1 | BRRI dhan28 | BRRI dhan67 | Binadhan-10 |
| **Grain quality** | | | | | | | |
| Amylose content | *Wx-A_group* | snpOS00445 (C) | C:C | C:C | C:C | C:C | C:C |
| | *Wx-GBSS-ex10* | snpOS00038 (T) | T:T | T:T | C:C | - | T:T |
| Chalkiness | *chalk5_576* | snpOS00024 (G) | - | G:G | A:A | G:G | A:A |
| Grain number | *Gn1a_1* | snpOS00396 (T) | T:T | T:T | A:A | T:T | A:A |
| **Abiotic stress** | | | | | | | |
| Salt-tolerance | *Saltol-Aus* | snpOS00397 (T) | - | - | G:G | - | G:G |
| | *Saltol-Aro* | snpOS00398 (T) | T:T | T:T | - | T:T | T:T |
| | *qSES1-2_2* | snpOS00409 (C) | - | - | T: T | - | C:C |
| | *qSES1-2_3* | snpOS00410 (A) | - | - | G:G | - | G:G |
| | *qSES1-2_4* | snpOS00411 (T) | | | A:A | T:T | T:T |
| Anaerobic germination | *qAG3_1* | snpOS00459 (C) | - | C:C | C:C | C:C | - |
| Cold tolerance | *qSCT1_1* | snpOS00403 (A) | - | A:A | - | - | - |
| **Biotic stress** | | | | | | | |
| Blast | *Pi-ta* | snpOS00006 (C) | A:A | C:C | A:A | C:C | A:A |
| | *Pb1* | snpOS00478 (T) | C:C | C:C | C:C | - | C:C |
| | *Pi33_1* | snpOS00468 (T) | G:G | G:G | G:G | - | G:G |
| | *Pi9_1* | snpOS00451 (C) | G:G | G:G | G:G | - | G:G |
| | *xa5-S1_SKEP* | snpOS00054 (AG) | - | - | TC:TC | - | TC:TC |
| BLB | *xa13_1* | snpOS00493 (C) | G:G | G:G | G:G | - | G:G |
| | *Xa21_SKEP* | snpOS00061 (C) | G:G | G:G | G:G | - | G:G |
| | *BPH17_3* | snpOS00430 (G) | A:A | G:G | A:A | - | A:A |
| BPH | *BPH32* | snpOS00442 (G) | C:C | C:C | G:G | - | C:C |
| | *BPH9* | snpOS00486 (A) | - | - | C:C | A:A | A:A |
| Galmidge | *Gm4_3* | snpOS00466 (A) | A:A | G:G | G:G | - | G:G |
| | *Gm4_4* | snpOS00467 (C) | G:G | G:G | G:G | - | C:C |
| Number of desirable traits present in each genotype | | | 4 | 8 | 3 | 7 | 4 |

**Fig 7. The illustrative view of grain quality properties of the studied new and check varieties.** The breeding lines IR83484-3-B-7-1-1-1 (BRRI dhan97), HHZ5-DT20-DT2-DT1 (BRRI dhan99); the checks BRRI dhan67, and BRRI dhan28 are displayed in the figure as paddy, parboiled, unparboiled, and cooked rice, respectively. Parboiled rice indicates partially boiling the rice within its husk, whereas unparboiled rice was dehusked without boiling.

(A), Gm4_4: snpOS00467 (C) for gall midge] from the Intertek (https://www.intertek.com/agriculture/agritech/ (accessed on 28 January 2023) outsourcing provider. Based on SNP data, HHZ5-DT20-DT2-DT1 reveal 8 QTLs and BRRI dhan67 concealed 7 QTLs, IR83484-3-B-7-1-1-1 and Binadhan-10 carried 4 QTLs and BRRI dhan28 bear 3 QTL that regulates the trait of curiosity. The proposed lines were categorized utilizing trait-specific SNP QTL showing the valuable traits with conforming favorable alleles connected to the trait-based SNP in Table 9. Meanwhile, the current study is on SNP-specific marker-assisted selection therefore we have

Table 10. Physico-chemical properties of the IR83484-3-B-7-1-1-1 and HHZ5-DT20-DT2-DT1.

| Tested entries | HRY | MO | MRL | MRB | LBR | TGW | ASV | AML | PT | CT | ER | IR |
|---|---|---|---|---|---|---|---|---|---|---|---|---|
| IR83484-3-B-7-1-1-1 | 63.2±0.87a | 70.1±2.05a | 5.6 ±0.66a | 2.5 ±0.14a | 2.3 ±0.14b | 25.5 ±1.04a | 7.0 ±0.14a | 25.4 ±0.28bc | 8.6 ±0.16a | 16.2 ±0.26b | 1.9 ±0.17a | 2.7 ±0.12a |
| HHZ5-DT20-DT2-DT1 | 61.7 ±0.58ab | 67.7±0.87b | 6.3 ±0.54a | 2.1 ±0.17a | 3.1 ±0.17a | 22.8 ±0.54b | 7.0 ±0.17a | 27.1 ±0.46ab | 7.9 ±0.11a | 14.2 ±0.40c | 1.5 ±0.12a | 3.1 ±0.20a |
| BRRI dhan28 (Sus. Ck.) | 63.5±1.04a | 70.1±2.06a | 5.8 ±0.23a | 1.9 ±0.14a | 3.1 ±0.32a | 22.7 ±0.61b | 6.0 ±0.20b | 28.0±0.34a | 8.5 ±0.17a | 18.3 ±0.11a | 1.5 ±0.17a | 4.0 ±0.14a |
| BRRI dhan67 (Tol. Ck.) | 61.1±0.85b | 69.4 ±1.85ab | 5.9 ±0.26a | 2.1 ±0.20a | 2.9 ±0.26a | 22.1 ±0.46b | 7.1 ±0.18a | 24.6±0.18c | 8.8 ±0.20a | 16.3 ±0.21b | 1.3 ±0.11a | 2.7 ±0.11a |

Three replications are used to display mean± SE in the data. Different lettering among the same columns implies a significant difference utilizing the p < 0.05 value. SE = Standard deviation. Here, HRY = Head rice yield (%), MO = Milling outturn (%), MRL = Milled rice length (mm), MRB = Milled rice breadth (mm), LBR = L/B ratio, TGW = Thousand grain weight (g), ASV = Alkali spreading value (ASV), AML = Amylose content (%), PT = Protein (%), CT = Cooking time (min), ER = Elongation ratio, IR = Imbibition ratio.

utilized only 20 gene-based markers. These 20-trait based QTLs are linked to specific SNPs with relevant desirable traits (traits of interest). These trait-connected markers are imposed to identify the basis of the studied genotype on the accessibility of valuable traits. The unrooted neighbor-joining tree and an UPGMA (unweighted pair group method with arithmetic mean) cluster dendrogram showing genetic relationships between the two proposed lines with different check varieties (S3 Fig) based on the 1K Rice Custom Amplicon assay or 1k-RiCA SNP markers. The proposed lines revealed distinct molecular variations/differences among with checks.

## 4. Discussions

The higher yield throughout the dry or *Boro* season is critical for rice cultivation in Bangladesh. The salinity of the soil and water is the most critical environmental factor, which hampers the normal growth and development of plants and there is a clear negative association between salinity and yield [17, 38]. In the regional yield trial, we found lower salinity levels (EC: 3.73–5.25 dS/m at Assasuni; EC: 2.47–6.99 dS/m at Debhata) and medium to high saline (EC: 3.96–15.01 dS/m at Kaliganj; EC: 7.50–13.20 dS/m at Koyra). The salinity ranges and status of the different trials and studied locations have given (Table 1 and S3 Table). Precise assessment and adaptability in the hotspot regions were suggested to assess the suitability of tested genotypes throughout the regions [38]. Since its release in 1994, BRRI dhan28 is a very popular dry season variety in the whole of Bangladesh. In saline water-induced areas in the southern part of the country, where BRRI dhan28 is grown as a short-duration variety with better grain parameters (i.e. medium slender grains with tastiness), high amylose content, high marketing value, and always demandable. Interestingly, BRRI dhan28, a very sensitive variety to salinity, can provide better yield in salt-affected coastal regions through partial escaping of medium to high salt stress during vegetative to flowering stages with the early maturing ability.

In regional yield trial noticed the most promising breeding lines that produced higher yield with higher level of salinity tolerance compared to susceptible check BRRI dhan28 and other tolerant check BRRI dhan67 and Binadhan-10. The IR83484-3-B-7-1-1-1, HHZ12-SA-L2-Y3-Y2 and HHZ5-DT20-DT2-DT1 showed better yield performance compare to checks. These lines were generated at the International Rice Research Institute (IRRI) by hybridizing IRRI113/BRRI dhan40, Huang-Hua-Zhan/Teqing, and Huang-Hua-Zhan/OM1723, respectively, using the pedigree selection technique. The IR83484-3-B-7-1-1-1 genotype is easily

recognized through anthocyanin coloration in the base of the plants and stems also. The distinguish characterization of Distinctness, Uniformity, and Stability (DUS) characters was represented in S2 Table. It is important to take into account genotypes as well as environmental interactions when making decisions about selection for yield improvement. Stability is the essential criterion for selection with a higher grain yield advantage [39, 40]. According to [29], the ASV stability test should be performed, due to knowing the adaptability and suitability of the tested genotypes, specific information was derivative from the AMMI score. The smaller ASVs give higher stability and indicate the integration of higher yield, which shows in bread wheat [41]. The higher ASVs with better yield performance were suggested for explicit adaptableness in finger millet for evaluation in various locations [42].

The AMMI results showed that genotypic effects, followed by GEI effects and environmental impacts, contributed the most. The higher value in IPCA 1 and IPCA 2 was enough to analyze the whole G-E interaction [43–45]. The existence of a substantial percentage of LGI dictates the analysis of the stability of elite rice genotypes over locations. This is consistent with [46, 47] findings on the various crops. The significant variation among the locations is an outcome of intrinsic variations in the environmental situations and influence that the studied locations were diversified. Based on yield effectiveness, [48] categorized productive locations into several agro-ecological areas, and these effective areas differ in soil qualities, rainfall, and temperatures, resulting in substantial GE. The significant error variance components and GE observed in this analysis provide obstacles in the selection of suited genotypes and breeding because they impede study repeatability, halting breeding and genotype selection efforts [49]. The larger GE and error variance components, according to [50], increase the cost of assessment because more replications, locations, and alike years are required to enhance heritability and, as a result, selection efficiency. The presence of considerable GE, as well as its variance component, which more than doubled the variance component for genotypes, necessitates the development of strategies to cope with it. Bernardo [51] identified three methods for dealing with severe GE: exploiting, avoiding, and mitigating. When breeders look for genotypes that are high yielding and stable, whereas when breeding for GE, breeders stratify habitats into more homogenous mega-environments with little GE. The which-won-where pattern and mega-environment delineation are valid if they are reproducible over seasons or years [52].

For investigating the practicable presence of crop varieties in different locations in the target environment, the image view of "which-won-where" of the multi-environment trial (MET) data is indispensable (Fig 4) [53, 54]. The best method for predicting the interactions between genotypes and environments, and for accurately predicting a biplot, is to use the polygon view of a biplot [55, 56]. In this investigation, the vertex genotypes are G4, G10, G5, and G9. The highest yield for the environments was given to the vertex genotype within the sector. Therefore, two mega-settings are produced in Fig 4 based on the biplot analysis data of three environments. The first mega-environment encircles the winner environments of Assasuni (E1) and Koyra (E3) with genotype G4 (IR83484-3-B-7-1-1-1); the second mega-environment encloses the winner environments of Debhata (E2) with genotype G10 (BR8992-B-18-2-26). The other winner genotypes G5 (IR87872-7-1-1-2-1-B) and G9 (BR8980-B-1-3-5) make up another mega-environment. Also, the genotype in the polygon (for example G3, G6, and G13 for Mega-E1) was less sensitive to the location than the winning genotype [45, 57]. The two other corner genotypes, G5 and G9, were low-yielding (Fig 4). They were situated so far from all of the test locations, recollecting the fact that they yielded lower at each location [57].

Rice expands more and becomes flakier as the amylose concentration rises. The amylose concentration of the rice can have a big impact on a lot of the cooking and eating qualities of milled rice [58]. Rice that has been cooked either softly or hard has an amylose concentration higher than 25%. Rice with a high amylose content is known to cook up dry and fluffy but

might harden after chilling [59]. Most rice-growing regions throughout the world favor rice with an intermediate amylose level because it produces soft, somewhat wet-cooked rice [60]. Rice is nearly instantly consumed after cooking, thus cooking it for a shorter amount of time might be advantageous, especially if fuel conservation is an issue. Cooking qualities have a significant influence on rice consumption preferences [13]. The short cooking time is observed in HHZ5-DT20-DT2-DT1. For cooked rice, the elongation ratio is a crucial factor. When rice lengthens more, it appears finer, and when rice girths out, it appears coarser [60]. The breeding line HHZ5-DT20-DT2-DT1 will be more acceptable in the greater Khulna, Bagerhat, and Satkhira districts for its long-slender grains. The grain size and shape of IR83484-3-B-7-1-1-1 was medium bold and it will be more popular in greater Barishal, Barguna, Patuakhali, and Pirojpur where farmers and consumers prefer the coarse rice. High-volume expansion is positively correlated with amylose content when the volume of cooked rice is ingested compared to the volume of uncooked rice [30]. The imbibition ratio was the highest in BRRI dhan28. However, plant breeders must create genotypes that are not only stable and high-yielding but also more acceptable grain quality parameters in terms of high amylose percentage, export-quality grain, and short duration. In this sense, combinedly considering the agronomic parameters and physicochemical characteristics may enable us to make decisions for selecting the genotypes with superior grain quality.

The proposed line IR83484-3-B-7-1-1-1 carried the four QTLs having high amylose content, high grain number, gall midge resistance as well as salt tolerance capacity. On the contrary, HHZ5-DT20-DT2-DT1 possessed 8 important QTLs having abiotic stress (salt and cold tolerance, anaerobic germination), biotic stress (blast, *BPH17*), and grain qualities parameters (amylose content, chalkiness, and grain number). The *Saltol_Aro* QTL is well-validated for the seedling stage salinity tolerance in Bangladesh. These QTLs are found in both proposed lines and the tolerant checks (BRRI dhan67 and Bina dhan-10). These were the best genotypes in terms of yield, some agronomic parameters (plant height, growth duration), and grain quality aspects as well as valuable traits that are directly and indirectly involved in crop growth and development. Finally, the 103th National Seed Board meeting (Bangladesh's authority for variety release on 8 September 2020, and the governmental gazette was published on 25 August 2021) released BRRI dhan97 and BRRI dhan99 for commercial cultivation as high salinity-tolerant rice varieties with simultaneous insurance for yield, grain qualities, and palatability of coastal farmers and consumers.

## 5. Conclusions

The current study used the AMMI stability model to analyze the fifteen salinity-tolerant elite breeding genotypes spread across various locations during three consecutive years. We found that the locations under study contributed significantly to one another in illuminating the overall variation in grain output. The eight genotypes including IR83484-3-B-7-1-1-1, IR87870-6-1-1-1-1-B, BR8992-B-18-2-26, HHZ5-DT20-DT2-DT1, HHZ12-SAL2-Y3-Y2, BR8980-B-1-3-5, BRRI dhan67, and Binadhan-10 have positive estimated breeding values which are used for parent selection. The three genotypes IR83484-3-B-7-1-1-1, HHZ5-DT20-DT2-DT1, and HHZ12-SAL2-Y3-Y2 were the best and selectable one based on the farmer's acceptability and good grain quality. Based on yield performance and salinity, these three genotypes revealed significantly higher yields compared to check varieties. SNP-specific categorization exposed that HHZ5-DT20-DT2-DT1 harbored the maximum effective QTLs (8) compare to BRRI dhan67 (7 QTLs) accountable for high amylose content, chalkiness, higher grain number, overall phenotypic performance/salt injury score, anaerobic germination, blast, and cold resistance. On the other hand, IR83484-3-B-7-1-1-1 and Binadhan-10

have 4 QTLs/genes that regulate the valuable traits. Nevertheless, these 20 vital SNPs are supportive of detecting the best genotypes for releasing new rice varieties for farmers and consumers. Finally, the top performer genotypes IR83484-3-B-7-1-1-1 and HHZ5-DT20-DT2-DT1 have been certified BRRI dhan97 and BRRI dhan99 as salinity tolerant varieties respectively, as well as could be used for large-scale commercial cultivation throughout saline prone areas. It is expected that those rice varieties will alleviate poverty in the saline-prone southern region of Bangladesh and hence will ensure sustainable rice production as well as food security.

## Supporting information

**S1 Fig. Correlation yield performance of three locations in the Regional Yield Trial (RYT) during Boro 2016–17.**
(PDF)

**S2 Fig. The salinity level of the proposed variety trial varied at different locations of the coastal zone during the proposed variety trial during boro2018-19.** The degree of salinity was found highest at Kaliganj and the lowest at Tala, Satkhira in farmers' fields.
(PDF)

**S3 Fig.** a) Unrooted neighbour-joining tree and b) an UPGMA (unweighted pair group method with arithmetic mean) cluster dendrogram showing genetic relationships between the four proposed lines (IR83484-3-B-7-1-1-1: BRRI dhan97, BR9011-67-4-1: BRRI dhan98, HHZ5-DT20-DT2-DT1: BRRI dhan99, HHZ12-SAL2-Y3-Y2 and three check varieties (BR26, BRRI dhan28, and BRRI dhan67) based on the 1K Rice Custom Amplicon assay or 1k-RiCA SNP markers. All four lines and three checks showed distinct molecular variations/differences among them.
(PDF)

**S4 Fig. Correlation between yield and yield-associated agronomic traits in the regional yield trial.** YLD: Yield, DM: Days to maturity, PH: Plant height, ET: Effective tillers, PL: Panicle length, UFG: Unfilled grain.
(PDF)

**S1 Table. Participatory varietal selection of rice genotypes grown at Assasuni, Debhata, Kaliganj in Satkhira and Koyra in Khulna districts during Boro2016-17.**
(PDF)

**S2 Table. Distinguish characterization of distinctness, uniformity and stability (DUS) test for BRRI dhan97 and BRRI dhan99 based on zadoks scale (Zadoks et al. 1974).**
(PDF)

**S3 Table. Soli salinity classes and the ranges of electrical conductivity (EC) (Source: Soil resource development institute, 2010).**
(PDF)

## Acknowledgments

The authors acknowledge the technical support provided by the plant breeding division, Grain quality and nutrition division, and adaptive research division of BRRI. We thank the International Rice Research Institute (IRRI) for sharing the germplasm.

## Author Contributions

**Conceptualization:** M. Akhlasur Rahman.

**Data curation:** Sanjoy K. Debsharma, M. Akhlasur Rahman, Mahmuda Khatun, Nusrat Jahan, Md. Ruhul Quddus, Sharifa S. Dipti.

**Formal analysis:** Sanjoy K. Debsharma, M. Akhlasur Rahman, Ribed F. Disha, Md. Ruhul Quddus, Hasina Khatun.

**Investigation:** Sanjoy K. Debsharma, M. Akhlasur Rahman, Mahmuda Khatun, Ribed F. Disha, Md. Ruhul Quddus, Md. Ibrahim.

**Methodology:** M. Akhlasur Rahman, Mahmuda Khatun, Sharifa S. Dipti.

**Project administration:** M. Akhlasur Rahman, Mahmuda Khatun.

**Resources:** M. Akhlasur Rahman, K. M. Iftekharuddaula, Md. Shahjahan Kabir.

**Software:** Sanjoy K. Debsharma, Hasina Khatun.

**Supervision:** M. Akhlasur Rahman, Mahmuda Khatun.

**Validation:** M. Akhlasur Rahman, Mahmuda Khatun, Hasina Khatun, Sharifa S. Dipti.

**Visualization:** Sanjoy K. Debsharma, Md. Ruhul Quddus.

**Writing – original draft:** Sanjoy K. Debsharma, M. Akhlasur Rahman, Ribed F. Disha.

**Writing – review & editing:** Sanjoy K. Debsharma, M. Akhlasur Rahman, Nusrat Jahan, K. M. Iftekharuddaula, Md. Shahjahan Kabir.

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
