## [Decision Letter · Decision Letter 0]

4 Jun 2023

PONE-D-23-12677Developing
climate-resilient rice varieties (BRRI dhan97 and BRRI dhan99) suitable for salt
stress environments in BangladeshPLOS ONE

Dear Dr. Rahman,

Thank you for submitting your manuscript to PLOS ONE. I have completed the evaluation
of the manuscript. After careful consideration, and reviewers suggestion, i feel
that it has merit but does not fully meet PLOS ONE’s publication criteria as it
currently stands. Therefore, I would like invite you to submit a revised version
followed my major revision of the manuscript that addresses the points raised during
the review process.

Please submit your revised manuscript by Jul 19 2023 11:59PM. If you will need more
time than this to complete your revisions, please reply to this message or contact
the journal office at plosone@plos.org. When
you're ready to submit your revision, log on to https://www.editorialmanager.com/pone/ and select the 'Submissions
Needing Revision' folder to locate your manuscript file.

Please include the following items when submitting your revised
manuscript:A rebuttal letter that responds to each point raised by the academic
editor and reviewer(s). You should upload this letter as a separate file
labeled 'Response to Reviewers'.A marked-up copy of your manuscript that highlights changes made to the
original version. You should upload this as a separate file labeled
'Revised Manuscript with Track Changes'.An unmarked version of your revised paper without tracked changes. You
should upload this as a separate file labeled 'Manuscript'.

If you would like to make changes to your financial disclosure, please include your
updated statement in your cover letter. Guidelines for resubmitting your figure
files are available below the reviewer comments at the end of this letter.

We look forward to receiving your revised manuscript.

Kind regards,

Md Ashrafuzzaman, Ph.D.

Academic Editor

PLOS ONE

6. Please include a separate caption for each figure in your manuscript.

7. We note that Figure 1 in your submission contain [map/satellite] images which may
be copyrighted. All PLOS content is published under the Creative Commons Attribution
License (CC BY 4.0), which means that the manuscript, images, and Supporting
Information files will be freely available online, and any third party is permitted
to access, download, copy, distribute, and use these materials in any way, even
commercially, with proper attribution. For these reasons, we cannot publish
previously copyrighted maps or satellite images created using proprietary data, such
as Google software (Google Maps, Street View, and Earth). For more information, see
our copyright guidelines: http://journals.plos.org/plosone/s/licenses-and-copyright.

8. We note that Figures 4, 5 and 6 in your submission contain copyrighted images. All
PLOS content is published under the Creative Commons Attribution License (CC BY
4.0), which means that the manuscript, images, and Supporting Information files will
be freely available online, and any third party is permitted to access, download,
copy, distribute, and use these materials in any way, even commercially, with proper
attribution. For more information, see our copyright guidelines: http://journals.plos.org/plosone/s/licenses-and-copyright.

a. You may seek permission from the original copyright holder of Figures 4, 5 and 6
to publish the content specifically under the CC BY 4.0 license.

b.If you are unable to obtain permission from the original copyright holder to
publish these figures under the CC BY 4.0 license or if the copyright holder’s
requirements are incompatible with the CC BY 4.0 license, please either i) remove
the figure or ii) supply a replacement figure that complies with the CC BY 4.0
license. Please check copyright information on all replacement figures and update
the figure caption with source information. If applicable, please specify in the
figure caption text when a figure is similar but not identical to the original image
and is therefore for illustrative purposes only.

Reviewers' comments:

Reviewer's Responses to Questions

**Comments to the Author**

1. Is the manuscript technically sound, and do the data support the conclusions?

Reviewer #1: Yes

Reviewer #2: Partly

2. Has the statistical analysis been performed
appropriately and rigorously? 

Reviewer #1: Yes

Reviewer #2: Yes

3. Have the authors made all data underlying the
findings in their manuscript fully available?

Reviewer #1: Yes

Reviewer #2: Yes

4. Is the manuscript presented in an intelligible
fashion and written in standard English?

Reviewer #1: Yes

Reviewer #2: No

5. Review Comments to the Author

Reviewer #1: The study on Developing climate-resilient rice varieties (BRRI dhan97
and BRRI dhan99) suitable for salt stress environments in Bangladesh is worthy of
investigation and reporting in understanding further this complex abiotic stress
affecting rice productivity severely. Study has successfully released two high
yielding rice cultivars for salinity prone areas in Bangladesh. However, there are
major corrections and supporting data is needed to validate the current study. The
comments are appended below:

1- Authors has tested a set of lines in salinity 'hotspot' locations with tolerant
and susceptible checks but not provided any yield data of these lines under non
stress/ normal conditions. Its very important that line/variety not only perform
under stress but also should perform high yielding with acceptable grain quality
under non stress conditions.

2- Salinity tolerance in rice is a multifactorial trait highly influenced by
environment, correlation with other traits is required to understand how agronomic
traits are affecting the yield under salinity.

3- Authors has not provided how breeding values are estimated? It could be explained
clearly which lines has shown high breeding value and what basis?

4- Authors has genotyped the lines/ varieties using 20 trait linked SNPs ad mostly
these are grain quality traits but what about screening with SNPs linked with
seedling stage/ reproductive stage SNPs for salinity? It will be crucial to see the
genotyping result for salinity traits also. Is SNPs for salinity traits not
available in rice? Include this also in discussion section.

5- What is the year of release of two salinity tolerant rice variety mentioned BRRI
dhan97 and BRRI dhan99? Its important to mention in manuscript either it’s a newly
released varieties or released long back if so mention the adoption rate in farmers
field.

Reviewer #2: The paper is very interesting and good piece of work from shortlisting
of elite lines to release varieties for commercial cultivation. However i found
certain issues to be improved for its scientific accuracy and recommendations. There
is no line number on MS, so difficult to prompt for specific part of MS.

1. There is no clue - how these 12 rice genotypes were selected? What was the basis
or past research efforts to choose these 12 genotypes? Arbitrary?

2. Please try to avoid grey sources to cite for big estimates. Do you feel that
reference 11 is good enough (working on PGP) to refer as prediction for 2050? Even
the reference 12 is a compilation from FAO and other sources. Please doubly check if
the narrative is correct. Once satisfied for authenticity of source, then cite. Same
issue for reference 4 : "billions of hectares are effected by salinity? Is this
correct? How many billion ha?; what percent of total arable area? You mentioned rice
is being cultivated in 166 mha globally (not billion), how much of that under
salinity?

3. Do you mean salt-responsive as quickly negatively affected by stress? Please
clarify.

4. Locations for RYT, ALART and PVT etc could be given in table along with stress
ranges to understand the locations easily. (M & M 2.1)

5. Forty to 45 days old seedlings is recommended practice for transplanting in the
saline areas? Similarly Zn dose is also lower than usual recommendation (5kg/ha). Is
this Bangladesh specific recommendation?

6. ASV doesn't give exact picture. A good breeding program should look for the YSI
(Yield stability index which encompass both stability (ASV) and yield ranking). This
can be incorporated quickly as it is a derived parameter but improve and strengthen
results for inferences. Please calculate and infer based on YSI. This will improve
the confusion on yield data vis-a-vis ASV on page 8.

7. In result table 1, 5 and 6; it would be good if authors could add the salinity
stress level against the locations to get the inferences from table immediately.
This will remove the confusion to say specific site or trial as stressed, for
example selected two genotypes yielded 6.24, 6.34 and 6.36 t/ha comparing to
tolerant and sensitive checks (6.41 and 6.01 t/ha respectively). [6 t/ha yield under
stress is good yield] - page 12. Two of them are having potential to produce
>7t/ha under favourable - Page 13. How much BR67 and and BR28 under favourable
envt?

8. Page 13 : 2 selected genotypes tolerates 14 dS/m salinity: for how many days
comparing to BR67?

9. In fig 3, G10 and G4 are on vertices and on positive side of PC1. But you selected
G11 and G4 (table 3). G11 is not shown as winning genotype. Could it be explained
please?

6. PLOS authors have the option to publish the peer
review history of their article (what does this mean?). If published, this will
include your full peer review and any attached files.

If you choose “no”, your identity will remain anonymous but your review may still be
made public.

**Do you want your identity to be public for this peer review?** For
information about this choice, including consent withdrawal, please see our
Privacy Policy.

Reviewer #1: **Yes: **SHAILESH YADAV

Reviewer #2: No

---

## [Author Response · Author response to Decision Letter 0]

5 Jul 2023

Date: 5 July 2023

To

Md Ashrafuzzaman, Ph.D.

Academic Editor

PLOS ONE

Ref: PONE-D-23-12677

Subject: Submission of a revised version of the manuscript: ‘Developing
climate-resilient rice varieties (BRRI dhan97 and BRRI dhan99) suitable for salt
stress environments in Bangladesh’ for evaluation and possible publication in the
journal ‘PLOS ONE’

Thank you so much for sending the reviewers’ comments for the manuscript: ‘Developing
climate-resilient rice varieties (BRRI dhan97 and BRRI dhan99) suitable for salt
stress environments in Bangladesh’ which has allowed us for considerable improvement
of the manuscript before publishing online in ‘PLOS ONE’. We are happy to inform you
that we have been able to address all of the reviewers’ comments. For clarity, the
original comments and suggestions made by the reviewers will appear in black colored
text, while our response will appear in blue text. Please note that all edits are
shown in track change in the manuscript. If unintentionally we have overlooked any
issue, kindly let us know and we will rectify that. 

We also confirm that all authors of the manuscript have read and approved the
submission of the revised version to ‘PLOS ONE. 

We hope that the manuscript is now in an acceptable form for publishing in ‘PLOS
ONE’. 

Please find below our responses to reviewers’ comments:

Reviewer#1: Comments and Suggestions for Authors

Reviewer #1: The study on Developing climate-resilient rice varieties (BRRI dhan97
and BRRI dhan99) suitable for salt-stress environments in Bangladesh is worthy of
investigation and reporting in understanding further this complex abiotic stress
affecting rice productivity severely. The study has successfully released two
high-yielding rice cultivars for salinity-prone areas in Bangladesh. However, there
are major corrections and supporting data is needed to validate the current study.
The comments are appended below:

Q1: Authors have tested a set of lines in salinity 'hotspot' locations with tolerant
and susceptible checks but have not provided any yield data of these lines under
non-stress/ normal conditions. It’s very important that line/variety not only
perform under stress but also should perform high yielding with acceptable grain
quality under non-stress conditions.

Response: Thank you for the question. We have conducted the regional yield trial
(RYT) in a non-stress condition in the BRRI Gazipur location. Also, the experiment
was performed in the ALART under non-stress conditions in the Gazipur location. 

Table: Performance of genotypes from Regional Yield Trial (RYT), Boro 2016-17,
Gazipur, Assasuni, Debhata, Kaliganj upazila under Satkhira district & Koyra
under Khulna district

SN

 Genotype GD

(Days) PH (cm) No. of Tiller Yield (t/ha)

 Assa Koyra *Gaz Kali Deb Mean

1 BR8940-B-17-4-7 142 85 9.99 6.15 4.89 5.47 0.0 5.05 5.39

2 BR8943-B-20-9-22 144 82 10.26 5.73 4.40 4.48 0.0 4.50 4.78

3 IR86385-85-2-1-B 145 96 10.3 5.66 5.57 4.64 0.0 3.74 4.90

4 IR83484-3-B-7-1-1-1 145 84 10.5 5.61 6.33 6.56 1.96 6.37 6.22

5 IR 87872-7-1-1-2-1-B 143 85 10.41 5.36 4.67 5.57 0.0 5.25 5.21

6 IR86385-117-1-1-B 146 82 10.34 5.50 4.78 4.57 0.0 4.87 4.93

7 IR 87870-6-1-1-1-1-B 149 91 10.0 6.37 5.73 5.46 0.76 5.69 5.81

8 BR8980-4-6-5 140 81 10.3 5.54 4.80 5.55 0.0 4.88 5.19

9 BR8980-B-1-3-5 142 78 10.34 5.60 5.51 4.49 0.0 5.56 5.29

10 BR8992-B-18-2-26 144 80 10.14 5.85 5.73 5.40 0.0 5.77 5.69

11 HHZ12-SAL2-Y3-Y2 143 100 10.32 6.53 6.50 6.36 0.96 6.12 6.38

12 HHZ5-DT20-DT2- DT1 143 91 10.25 5.80 6.47 6.61 1.32 6.18 6.27

13 BRRI dhan28 (S. Ck) 142 86 10.38 4.82 3.59 4.50 0.0 4.73 4.41

14 BRRI dhan67 (Ck) 143 92 10.49 6.33 5.63 5.60 0.0 5.42 5.75

15 BINA dhan10 (Ck) 147 95 10.26 6.45 5.65 5.48 0.0 6.05 5.91

H2b

 0.62 0.67 0.33 0.55 0.58 0.45 0.80 0.89 

LSD0.05

 1.66 1.34 1.27 0.90 0.80 0.5 0.80 

CV (%)

 0.30 1.80 18.3 7.60 7.00 13.0 6.90 

*non-stress trial

Q2: Salinity tolerance in rice is a multifactorial trait highly influenced by
environment, correlation with other traits is required to understand how agronomic
traits are affecting the yield under salinity.

Response: Salinity is a complex trait that is highly affected by the environment, so
we estimated the site correlations of the different locations and depicted positive
correlations among the locations (See Figure 4). Also, we determined the agronomic
trait correlation using six characters in RYT (see Supplementary Figure 4
below).

Q3: Authors has not provided how breeding values are estimated? It could be explained
clearly which lines has shown the high breeding value and what basis?

Response: We determined the estimated breeding value (EBV) using R (version 3.5.1),
which is found in the Materials and Method (2.6 Estimation of breeding value)
section. We calculated the accuracy/reliability of EBV derived from prediction error
variance (PEV) as the square root of 1 minus the ratio between PEV and additive
genetic variance (VA) i.e., rel < ‒sqrt(1-PEV/VA). The higher EBV was found in
G12: HHZ12-SAL2-Y3-Y2 (0.70 tha-1) and the lower one was for G1: BR8940-B-17-4-7 (-
0.12 tha-1).

Q4: Authors have genotyped the lines/ varieties using 20 trait-linked SNPs and mostly
these are grain quality traits but what about screening with SNPs linked with
seedling stage/ reproductive stage SNPs for salinity? It will be crucial to see the
genotyping result for salinity traits also. Is SNPs for salinity traits not
available in rice? Include this also in the discussion section.

Response: We have genotyped the proposed lines and checks utilizing 20 trait-linked
SNPs for grain qualities (amylose, chalkiness, and grain number), abiotic stress
(salt & cold tolerance, and anaerobic germination), and biotic stress (Blast,
BLB, BPH, and gall midge). The Saltol_Aro QTL is well-validated for the seedling
stage salinity tolerance in Bangladesh. These QTLs are found in both proposed lines
and the tolerant checks (BRRI dhan67 and Bina dhan-10). In the discussion section,
it has been written as a text.

Q5: What is the year of release of the two salinity-tolerant rice varieties mentioned
BRRI dhan97 and BRRI dhan99? It’s important to mention in the manuscript either it’s
a newly released variety or released long back if so, mention the adoption rate in
farmers’ field.

Response: BRRI dhan97 and BRRI dhan99 were released on 8 September 2020 through the
103rd National Seed Board (NSB) meeting and the Governmental gazette was published
on 25 August 2021. It has been written in the revised manuscript in the discussion
part. The adoption rate is not yet estimated by the socioeconomic group of BRRI.

Reviewer: 2_Comments and Suggestions for Authors

Reviewer #2: The paper is very interesting and good piece of work from shortlisting
of elite lines to release varieties for commercial cultivation. However, I found
certain issues to be improved for its scientific accuracy and recommendations. There
is no line number on MS, so difficult to prompt for specific parts of MS.

Response: Thank you for the encouraging comments and great suggestions.

Q1. There is no clue - how these 12 rice genotypes were selected? What was the basis
or past research efforts to choose these 12 genotypes? Arbitrary?

Response: In the variety released system of Bangladesh, there are several steps to
release a rice variety: At first the genotypes were selected by the Pedigree method
(F2 to F6), then carried out Observational Yield Trial (OYT) followed by Preliminary
Yield Trial (PYT), Regional Yield Trial (RYT), Advanced Line Adaptive Research Trial
(ALART) and finally Proposed Variety Trial (PVT). The genotypes were selected based
on yield, growth duration, and plant height compare checks. Genotypes G3, G4, G5,
G6, G7, G11and G12 were introduced by International Rice Research Institute (IRRI),
Los Banos, Philippines.

Q2. Please try to avoid grey sources to cite for big estimates. Do you feel that
reference 11 is good enough (working on PGP) to refer to as prediction for 2050?
Even reference 12 is a compilation from FAO and other sources. Please doubly check
if the narrative is correct. Once satisfied for authenticity of source, then cite.
Same issue for reference 4: "billions of hectares are effected by salinity? Is this
correct? How many billion ha?; what percent of total arable area? You mentioned rice
is being cultivated in 166 mha globally (not billion), how much of that under
salinity?

Response: Of the total 14 billion ha of land available on earth, about 1 billion ha
are natural saline soils. Also, it is estimated that worldwide about 20% of total
cultivated lands and 33% of irrigated agricultural lands are afflicted by high
salinity (Epstein et al., 1980; Shrivasata and Kumar, 2015).

Epstein, E., Norlyn, J. D., Rush, D. W., Kingsbury, R. W., Kelly, D. B., Gunningham,
G. A. and Wrona, A. F. 1980. Saline culture of crops: A genetic approach. Science,
210: 399–404.

Shrivastava, P., & Kumar, R. (2015). Soil salinity: A serious environmental issue
and plant growth promoting bacteria as one of the tools for its alleviation. Saudi
journal of biological sciences, 22(2), 123-131.

It has been estimated that more than 50% of the arable land would be salinized by the
year 2050 (Jamil et al., 2011).

Jamil, A., S. Riaz, M. Ashraf, and M. R. Foolad. "Gene expression profiling of plants
under salt stress." Critical Reviews in Plant Sciences 30, no. 5 (2011):
435-458.

Q3. Do you mean salt-responsive as quickly negatively affected by stress? Please
clarify.

Response: Salt stress is a major environmental stress that affects plant growth and
development. Salt stress increases the intracellular osmotic pressure and can cause
the accumulation of sodium to toxic levels. Like other abiotic stresses, salt stress
negatively affects plant growth and reproduction in many ways. Being a
non-halophytic plant/glycophyte, the salt-responsiveness of rice is obviously
negative, and quick destruction of the plant tissue (beyond its threshold level) has
occurred during the seedling and reproductive stages.

Q4. Locations for RYT, ALART and PVT etc could be given in table along with stress
ranges to understand the locations easily. (M & M 2.1)

Response: The table contains trial, locations along with water salinity status given
below:

Experiment/Trial Location name Status of water salinity

RYT Assasuni, Sathkira Low stress 

 Kaliganj, Satkhira High stress

 Debhata, Satkhira Low to high stress

 Koyra, Khulna Medium stress

ALART BRRI Gazipur No stress (Favorable)

 Debhata, Satkhira Low to high stress

 Assasuni, Satkhira Favorable

 Batiaghata, Satkhira Low to medium stress

 Paikgacha, Khulna High stress

 Kalapara, Barguna Low stress

 Pathorghata, Barguna Low to medium stress

PVT Tala, Satkhira Medium stress

 Debhata, Satkhira Low to high stress

 Kaliganj, Satkhira High stress

 Dumuria, Khulna Low stress

 Paikgacha, Khulna High stress

 Batiaghata, Khulna Low to medium stress

 Rampal, Bagerhat Low stress

 Kalapara, Barguna Low to medium stress

Q5. Forty to 45 days old seedlings is recommended practice for transplanting in the
saline areas? Similarly, Zn dose is also lower than usual recommendation (5kg/ha).
Is this Bangladesh specific recommendation?

Response: In Bangladesh, there are three seasons: Aus, Aman and Boro. In Boro
seasons, 40 to 45 days-old-seedlings are transplanted in the main field in the
coastal saline areas as well. We followed the BRRI recommended doses according to
Agro-ecological Zones (AEZ) specificity in Bangladesh. Generally, 10-11 kg ZnSO4/ ha
was applied during boro season.

Q6. ASV doesn't give exact picture. A good breeding program should look for the YSI
(Yield stability index which encompass both stability (ASV) and yield ranking). This
can be incorporated quickly as it is a derived parameter but improve and strengthen
results for inferences. Please calculate and infer based on YSI. This will improve
the confusion on yield data vis-a-vis ASV on page 8.

Response: The lower value of YSI is considered as the most stable with a high grain
yield. In this case, G4 and G11 were the most stable genotypes.

Supplementary Table 3: The YSI (Yield stability index), ASV and yield ranking.

Geno code Mean Yield Rank (Yield) ASV Rank (ASV) YSI

G1 5.36 9 0.50 6 15

G2 4.88 14 0.57 5 19

G3 4.99 13 1.13 1 14

G4 6.1 3 0.87 2 5

G5 5.09 10 0.34 7 17

G6 5.05 12 0.15 14 26

G7 5.93 5 0.14 15 20

G8 5.07 11 0.16 13 24

G9 5.56 8 0.30 8 16

G10 5.78 7 0.26 10 17

G11 6.15 2 0.71 3 5

G12 6.38 1 0.26 11 12

G13 4.38 15 0.65 4 19

G14 5.8 6 0.27 9 15

G15 6.05 4 0.24 12 16

Q7. In result table 1, 5 and 6; it would be good if authors could add the salinity
stress level against the locations to get the inferences from table immediately.
This will remove the confusion to say specific site or trial as stressed, for
example selected two genotypes yielded 6.24, 6.34 and 6.36 t/ha comparing to
tolerant and sensitive checks (6.41 and 6.01 t/ha respectively). [6 t/ha yield under
stress is good yield] - page 12. Two of them are having the potential to produce
>7.0 t/ha under favorable - Page 13. How much BR67 and BR28 under favourable
envt?

Response: BRRI dhan28 (released in 1994) is a mega variety for Boro season with a
limitation of its susceptibility to blast disease and salinity. BRRI dhan67 is
another excellent salinity-tolerant variety but its salinity tolerance threshold
limit is 8 dS/m in the whole life cycle. Therefore, the development of new varieties
(BRRI dhan97 and BRRI dhan99) with more tolerance to salinity like 12 dS/m is a new
milestone achievement in salinity breeding.

Q8. Page 13: 2 selected genotypes tolerate 14 dS/m salinity: for how many days
comparing to BR67?

Response: The salinity level was observed 14 dS/m consistently at Kaliganj, Satkhira
where BRRI dhan97 and BRRI dhan99 were evaluated under PVT. In that situation, both
the selected genotypes had survived well and produced considerable yield but BRRI
dhan67 (Salt-tolerant check) was completely damaged within 7 days after
transplanting as its salinity tolerance level is 8 dS/m (see Figure 5).

Q9. In fig 3, G10 and G4 are on vertices and on positive side of PC1. But you
selected G11 and G4 (table 3). G11 is not shown as the winning genotype. Could it be
explained, please?

Response: As mentioned earlier, G11 was not the winning genotype in Figure 3, yet it
was the top most preferable genotype in the Participatory Varietal Selection (PVS)
approach by the farmers of Koyra, Khulna location due to its salinity tolerance,
grain quality parameters including long slender grain as well as yield
performance.

---

## [Decision Letter · Decision Letter 1]

19 Sep 2023

PONE-D-23-12677R1Developing
climate-resilient rice varieties (BRRI dhan97 and BRRI dhan99) suitable for salt
stress environments in BangladeshPLOS ONE

Dear Dr. Rahman,

Thank you for submitting your manuscript to PLOS ONE. After careful consideration and
based on the reviewers suggestions, I would like to invite you to submit a revised
version of the manuscript following minor revision that addresses the points raised
during the review process.

Please submit your revised manuscript by Nov 03 2023 11:59PM. If you will need more
time than this to complete your revisions, please reply to this message or contact
the journal office at plosone@plos.org. When
you're ready to submit your revision, log on to https://www.editorialmanager.com/pone/ and select the 'Submissions
Needing Revision' folder to locate your manuscript file.

Please include the following items when submitting your revised
manuscript:A rebuttal letter that responds to each point raised by the academic
editor and reviewer(s). You should upload this letter as a separate file
labeled 'Response to Reviewers'.A marked-up copy of your manuscript that highlights changes made to the
original version. You should upload this as a separate file labeled
'Revised Manuscript with Track Changes'.An unmarked version of your revised paper without tracked changes. You
should upload this as a separate file labeled 'Manuscript'.If you would like to make changes to your financial disclosure,
please include your updated statement in your cover letter. Guidelines for
resubmitting your figure files are available below the reviewer comments at the end
of this letter.

We look forward to receiving your revised manuscript.

Kind regards,

Md Ashrafuzzaman, Ph.D.

Academic Editor

PLOS ONE

Journal Requirements:

Reviewers' comments:

Reviewer's Responses to Questions

**Comments to the Author**

1. If the authors have adequately addressed your comments raised in a previous round
of review and you feel that this manuscript is now acceptable for publication, you
may indicate that here to bypass the “Comments to the Author” section, enter your
conflict of interest statement in the “Confidential to Editor” section, and submit
your "Accept" recommendation.

Reviewer #1: (No Response)

Reviewer #2: (No Response)

2. Is the manuscript technically sound, and do the data
support the conclusions?

Reviewer #1: Yes

Reviewer #2: Yes

3. Has the statistical analysis been performed
appropriately and rigorously? 

Reviewer #1: Yes

Reviewer #2: Yes

4. Have the authors made all data underlying the
findings in their manuscript fully available?

Reviewer #1: Yes

Reviewer #2: Yes

5. Is the manuscript presented in an intelligible
fashion and written in standard English?

Reviewer #1: Yes

Reviewer #2: Yes

6. Review Comments to the Author

Reviewer #1: I agree with authors response and changes made in this revised version.
Kindly include regional yield trial(RYT) data Table in supplementary section and
also include related texts in results section.

Reviewer #2: Authors tried to address the issues raised (reviewer 2) but not
addressed in the manuscript. I would like to reiterate that we need redressal of
points and their inclusion in the main manuscript.

1. We know the process of varietal release in Bangladesh but it is not known to all
the readers. So in the M & M, please include a passing remark in section 2.2
that best performing 12 genotypes are derived from OYT and PYT conducted at XYZ
location. If you can, please give the initial numbers of genotypes in OYT and PYT
mentioning that: original data on this is not reported, it is acceptable. This way,
readers will know that these 12 genotypes are developed through a fixed process (not
arbitrary).

2. When i mentioned that please avoid giving grey literature or not use unless very
authentic/related reference, authors responded with citation from authorities like
Epstein but not included into manuscript. Why? Just to convince me? We need to
convince readers. Can a paper on "Plant Growth Promoting" bacteria substantiate it,
that too a very low ranking journal? Probably authors could not understand what i
wanted to convey. Please do the needful.

3. About my query number 4 on stress status of trail locations. Again Authors tried
to respond me through a very vague table. It should be very precise or with range of
stress. What is low, medium, high, low to high ? Can authors mention some range
atleast and give as supplementary table to strengthen 2.1 section? It should be for
all readers (not only for me).

4. About my query 5: there is something wrong with Zn calculation. You mentioned
10-11 kg ZnSO4/ha. But the sentence in section 2.3 says: "Fertilizers were supplied
at a rate of 120:19:60: 20:3.6 kg NPKSZn/ha (260-97-120-110-11kg/ha, respectively in
the form of urea, TSP, MoP, gypsum, and zinc sulphate)". The minimum Zn is 20% in
ZnSO4 and once Zn is heptahydrate or so, the concentration varies. So 3.6 Zn cannot
be 11 kg ZnSO4, by any means. So please confirm if 3.6 is correct or 11 kg
correct.

5. About query 6 on YSI, i did not find any mention of YSI calculation in M & M.
Please include its derivation. When Fig 3 is not able to conclusively select G11;
and your selection is based on YSI for G4 and G11; why you donot include the YSI
supplementary table 3 as main table in the text. Please do so as this table is the
concrete basis of your genotype selection.

6. With reference to my query 7, you did not address what i wanted to do. But if you
don't want include the stress level in table 1, 5 and 6 (but it would be better if
you do); please respond to my above-mentioned query no.3 with definite range.

7. Few small corrections: under section 2.7 (Chemical property), another abbreviation
of ASV used. Please highlight this in section that it is alkali spreading value for
grain quality trait. Otherwise, reader may confuse. Another correction in table 3,
column 6, 7 and 8 are for grain yield at XYZ location. Please correct it.

7. PLOS authors have the option to publish the peer
review history of their article (what does this mean?). If published, this will
include your full peer review and any attached files.

If you choose “no”, your identity will remain anonymous but your review may still be
made public.

**Do you want your identity to be public for this peer review?** For
information about this choice, including consent withdrawal, please see our
Privacy Policy.

Reviewer #1: **Yes: **SHAILESH YADAV, Africa Rice Center (AfricaRice), Cote
d’Ivoire

Reviewer #2: No

---

## [Author Response · Author response to Decision Letter 1]

8 Oct 2023

Reviewer #1: I agree with author's response and changes made in this revised version.
Kindly include the regional yield trial (RYT) data Table in the supplementary
section and also include related texts in the results section.

Response: Thank you for the encouraging comments and great suggestions. In the
manuscript, the regional yield trial (RYT) data Table already exists in the Result
section (Table 4), and related texts are cited in the result and discussion section. 

Reviewer #2: Authors tried to address the issues raised (reviewer 2) but not
addressed in the manuscript. I would like to reiterate that we need redressal of
points and their inclusion in the main manuscript.

Response: Thank you for the encouraging comments and great suggestions.

Q1. We know the process of varietal release in Bangladesh but it is not known to all
the readers. So in the M & M, please include a passing remark in section 2.2
that best performing 12 genotypes are derived from OYT and PYT conducted at XYZ
location. If you can, please give the initial numbers of genotypes in OYT and PYT
mentioning that: original data on this is not reported, it is acceptable. This way,
readers will know that these 12 genotypes are developed through a fixed process (not
arbitrary).

Response: In the manuscript, we included the varietal release system in Bangladesh as
Table 1 and the background history of the studied genotypes in the M & M
(section 2.2) as well as OYT and PYT conducted at the different locations. 

Q2. When i mentioned that please avoid giving grey literature or not use it unless
very authentic/related reference, authors responded with citations from authorities
like Epstein but not included into the manuscript. Why? Just to convince me? We need
to convince readers. Can a paper on "Plant Growth Promoting" bacteria substantiate
it, that too a very low-ranking journal? Probably authors could not understand what
i wanted to convey. Please do the needful.

Response: We addressed your suggestions and cited relevant references in the
manuscript. 

Q3. About my query number 4 on stress status of trail locations. Again Authors tried
to respond me through a very vague table. It should be very precise or with a range
of stress. What is low, medium, high, low to high? Can authors mention some range at
least and give as supplementary table to strengthen 2.1 section? It should be for
all readers (not only for me).

Response: We made a table containing the trials, locations, salinity status, and
ranges (minimum to maximum). We attached this table in the main manuscript as a
Table 1. Besides, we also attached soli salinity classes and the ranges of
Electrical Conductivity (EC) as a Supplementary Table 3.

Table 1: The characterization of the studied areas including trials, locations,
salinity status, and salinity ranges (minimum to maximum) 

Trials Location name Status of salinity Ranges (dS/m)

RYT Assasuni, Sathkira Low stress (Non saline to very slight saline) 3.73-5.25

 Kaliganj, Satkhira High stress (very slight saline to moderate saline)
3.96-15.01

 Debhata, Satkhira Low to high stress (very slight saline to slight saline)
2.47-6.99

 Koyra, Khulna Medium stress (slight saline to moderate saline) 7.85-13.20

ALART BRRI Gazipur No stress (Favorable) 

 Debhata, Satkhira Low to high stress (very slight saline to slight saline)
3.25-7.09

 Assasuni, Satkhira Low stress (Non saline to very slight saline) 3.90-6.05

 Batiaghata, Satkhira Low to medium stress (very slight saline to slight saline)
2.70-7.35

 Paikgacha, Khulna High stress (very slight saline to moderate saline) 4.13-11.58

 Kalapara, Barguna Low stress (Very slight saline to slight saline) 3.53-7.01

 Pathorghata, Barguna Low to medium stress (very slight saline to slight saline)
3.21-7.83

PVT Tala, Satkhira Medium stress (Very slight saline to slight saline) 4.07-8.17

 Debhata, Satkhira Low to high stress (very slight saline to moderate saline)
3.10-12.29

 Kaliganj, Satkhira High stress (very slight saline to strong saline) 3.55-16.17

 Dumuria, Khulna Low stress (Very slight saline to slight saline) 3.61-6.79

 Paikgacha, Khulna High stress (Very slight saline to moderate saline) 4.23-11.79

 Batiaghata, Khulna Low to medium stress (very slight saline to slight saline)
3.40-8.50

 Rampal, Bagerhat Low stress (Very slight saline to slight saline) 4.08-7.53

 Kalapara, Barguna Low to medium stress (very slight saline to slight saline)
3.93-6.99

Supplementary Table 3: Soli salinity classes and the range of Electrical Conductivity
(EC) (Source: Soil Resource Development Institute, 2010)

Salinity class EC (dSm-1)

Non-saline to very slight saline 2-4

Very slight saline to slight saline 4-8

slight saline to moderate saline 8-12

Moderate saline to strong saline 12-16

strong saline to very strong saline >16

Q4. About my query 5: there is something wrong with Zn calculation. You mentioned
10-11 kg ZnSO4/ha. But the sentence in section 2.3 says: "Fertilizers were supplied
at a rate of 120:19:60: 20:3.6 kg NPKSZn/ha (260-97-120-110-11kg/ha, respectively in
the form of urea, TSP, MoP, gypsum, and zinc sulphate)". The minimum Zn is 20% in
ZnSO4 and once Zn is heptahydrate or so, the concentration varies. So 3.6 Zn cannot
be 11 kg ZnSO4, by any means. So please confirm if 3.6 is correct or 11 kg is
correct.

Response: 11 kg is correct and Zn content is also corrected in the section 2.3 of the
Manuscript "Fertilizers were supplied at a rate of 120:19:60: 20:4.01 kg NPKSZn/ha
(260-97-120-110-11kg/ha, respectively in the form of urea, TSP, MoP, gypsum, and
zinc sulphate)".

Q5. About query 6 on YSI, i did not find any mention of YSI calculation in M & M.
Please include its derivation. When Fig 3 is not able to conclusively select G11;
and your selection is based on YSI for G4 and G11; why you donot include the YSI
supplementary table 3 as main table in the text. Please do so as this table is the
concrete basis of your genotype selection.

Response: Yield stability index (YSI) derivation is attached in the M & M section
(2.5) and Yield Rank (RYi), ASV Rank (RASVi), and YSI value are included in the
manuscript as a table (Table 4).

Yield stability index (YSI) for individual genotypes which combines both mean grains
yield and ASV index was estimated as YSI = RASVi + RYi. Here, the rank of the AMMI
stability value for the ith genotype is RASVi, while the rank of the mean grain
yield for the ith genotype across environments is RYi.

Included in Table 4: The YSI (Yield stability index), ASV and yield ranking.

Geno code Mean Yield ASV Yield Rank (RYi) ASV Rank (RASVi) YSI

G1 5.36 0.50 9 6 15

G2 4.88 0.57 14 5 19

G3 4.99 1.13 13 1 14

G4 6.1 0.87 3 2 5

G5 5.09 0.34 10 7 17

G6 5.05 0.15 12 14 26

G7 5.93 0.14 5 15 20

G8 5.07 0.16 11 13 24

G9 5.56 0.30 8 8 16

G10 5.78 0.26 7 10 17

G11 6.15 0.71 2 3 5

G12 6.38 0.26 1 11 12

G13 4.38 0.65 15 4 19

G14 5.8 0.27 6 9 15

G15 6.05 0.24 4 12 16

Here, RASVi = Rank of the AMMI stability value for the ith genotype, RYi = Rank of
the mean grain yield for the ith genotype across environments

Q6. With reference to my query 7, you did not address what i wanted to do. But if you
don't want include the stress level in table 1, 5 and 6 (but it would be better if
you do); please respond to my above-mentioned query no.3 with definite range.

Response: We have already mentioned (query no.3) the salinity ranges and status of
the different trials and studied locations are given as a manuscript and
supplementary (Table1 & Supplementary Table 3).

Q7. Few small corrections: under section 2.7 (Chemical property), another
abbreviation of ASV used. Please highlight this in the section that it is an alkali
spreading value for grain quality trait. Otherwise, the reader may be confused.
Another correction in table 3, column 6, 7 and 8 are for grain yield at XYZ
location. Please correct it.

Response: We corrected the alkali spreading value (ASV) in section 2.7 for chemical
properties. Another in Table 4 is corrected by Assasuni (E1) yield (tha-1), Debhata
(E2) yield (tha-1) and Koyra (E3) yield (tha-1).

---

## [Decision Letter · Decision Letter 2]

5 Nov 2023

Developing climate-resilient rice varieties (BRRI dhan97 and BRRI dhan99) suitable
for salt stress environments in Bangladesh

PONE-D-23-12677R2

Dear Dr. Akhlasur Rahman,

I am pleased to inform you that your manuscript has been judged scientifically
suitable for publication and will be formally accepted for publication once it meets
all outstanding technical requirements.

Kind regards,

Md Ashrafuzzaman, Ph.D.

Academic Editor

PLOS ONE

Additional Editor Comments (optional):

Reviewers' comments:

Reviewer's Responses to Questions

**Comments to the Author**

1. If the authors have adequately addressed your comments raised in a previous round
of review and you feel that this manuscript is now acceptable for publication, you
may indicate that here to bypass the “Comments to the Author” section, enter your
conflict of interest statement in the “Confidential to Editor” section, and submit
your "Accept" recommendation.

Reviewer #2: All comments have been addressed

2. Is the manuscript technically sound, and do the data
support the conclusions?

Reviewer #2: Yes

3. Has the statistical analysis been performed
appropriately and rigorously? 

Reviewer #2: Yes

4. Have the authors made all data underlying the
findings in their manuscript fully available?

Reviewer #2: Yes

5. Is the manuscript presented in an intelligible
fashion and written in standard English?

Reviewer #2: Yes

6. Review Comments to the Author

Reviewer #2: Authors have addressed to the queries to my satisfaction, hence it is
ready for the publication in PLOS.

7. PLOS authors have the option to publish the peer
review history of their article (what does this mean?). If published, this will
include your full peer review and any attached files.

If you choose “no”, your identity will remain anonymous but your review may still be
made public.

**Do you want your identity to be public for this peer review?** For
information about this choice, including consent withdrawal, please see our
Privacy Policy.

Reviewer #2: **Yes: **Rakesh Kumar Singh, Principal Scientist and Program
Leader (CDG), ICBA Dubai

---

## [Editor Report · Acceptance letter]

10 Nov 2023

PONE-D-23-12677R2 

Developing climate-resilient rice varieties (BRRI dhan97 and BRRI dhan99) suitable
for salt-stress environments in Bangladesh 

Dear Dr. Rahman:

I'm pleased to inform you that your manuscript has been deemed suitable for
publication in PLOS ONE. Congratulations! Your manuscript is now with our production
department. 

Kind regards, 

on behalf of

Dr. Md Ashrafuzzaman 

Academic Editor

PLOS ONE